# Does drought advance the onset of autumn leaf senescence in temperate deciduous forest trees?

Bertold Mariën[1,*], Inge Dox[1], Hans J De Boeck[1], Patrick Willems[2], Sebastien Leys[1], Dimitri Papadimitriou[3] and Matteo Campioli[1]

[1]PLECO (Plants and Ecosystems), Department of Biology, University of Antwerp, 2160 Wilrijk, Belgium
[2]Hydraulics Division, KU Leuven, Kasteelpark Arenberg 40, 3001, Leuven, Belgium
[3]IDLab (Internet Data Lab), Department of Mathematics and Computer Science, University of Antwerp, 2000 Antwerp, Belgium

*Author for correspondence:
*Bertold Mariën*
*Tel: 032659333*
*Email: bertold.marien@uantwerpen.be*

## Abstract

- Severe droughts are expected to become more frequent and persistent. However, their effect on autumn leaf senescence, a key process for deciduous trees and ecosystem functioning, is currently unclear. We hypothesized that (I) severe drought advances the onset of autumn leaf senescence in temperate deciduous trees and that (II) tree species show different dynamics of autumn leaf senescence under drought.

- We tested these hypotheses using a manipulative experiment on beech saplings and three years of monitoring mature beech, birch and oak trees in Belgium. The autumn leaf senescence was derived from the seasonal pattern of the chlorophyll content index and the loss of canopy greenness using generalized additive models and piece-wise linear regressions.

- Drought and associated heat stress and increased atmospheric aridity did not affect the onset of autumn leaf senescence in both saplings and mature trees, even if the saplings showed a high mortality and the mature trees an advanced loss of canopy greenness. We did not observe major differences among species.

- Synthesis: The timing of autumn leaf senescence appears conservative across years and species, and even independent on drought, heat and increased atmospheric aridity. Therefore, to study autumn senescence and avoid confusion among studies, seasonal chlorophyll dynamics and loss of canopy greenness should be considered separately.

## Key words

Autumn leaf senescence, *Betula pendula*, Drought, Heat stress and increased atmospheric aridity , *Fagus sylvatica*, Generalized additive mixed models, Leaf coloration and fall, *Quercus robur*, Rainfall deficit

# 1. Introduction

Autumn leaf senescence is a developmental stage of the leaf cells. The core function of this process is the remobilization of nutrients and death is its consequence (Medawar, 1957;Keskitalo et al., 2005). Its evolutionary purpose is likely stress resistance and, as such, the process dynamics are affected by different forms of environmental stress (e.g. high temperatures, water logging) (Benbella and Paulsen, 1998;Leul and Zhou, 1998;Munné-Bosch and Alegre, 2004). The process of autumn leaf senescence is highly coordinated and characterized by a tight control over its timing. Furthermore, its most manifest feature, the detoxification of chlorophyll, allows the degradation of leaf macromolecules and subsequent nutrient remobilization -the essence of autumn leaf senescence- (Hörtensteiner and Feller, 2002;Munné-Bosch and Alegre, 2004;Matile, 2000). In addition, chlorophyll degradation allows for the typical leaf coloration during autumn. However, autumn leaf senescence is also an important process at the ecosystem scale because it affects multiple ecological processes, such as trophic dynamics, tree growth or the exchange of matter and energy between the ecosystem and atmosphere (Richardson et al., 2013).

Literature reports several definitions of autumn senescence and of multiple observational methods to measure autumn senescence (Gill et al., 2015;Fracheboud et al., 2009;Gallinat et al., 2015). This has hampered our understanding of the effects of drought stress on the timing of the onset of autumn leaf senescence, as opposed to the timing of leaf abscission or accelerated leaf senescence. For example, Estiarte and Penuelas (2015) reported that leaf senescence advances due to drought stress, while Vander Mijnsbrugge et al. (2016) reported a delay in the leaf senescence of young trees subjected to drought. After the summer drought in central Europe of 2003, Leuzinger et al. (2005) even reported that the leaf longevity (measured as a delay in the leaf discoloration and fall) of five deciduous tree species was on average prolonged by 22 days.

Droughts are expected to occur more frequently and become more intensive due to global warming and changes in precipitation patterns (IPCC, 2014;Crabbe et al., 2016). Extended periods with lower than average rainfall are often associated with higher air temperatures and higher vapor pressure deficits, which can negatively affect the functioning of trees in the temperate zone (Novick et al., 2016;De Boeck and Verbeeck, 2011). Belgian forests are thought to be especially vulnerable to droughts as they typically have sandy soils with low soil field capacities (Vander Mijnsbrugge et al., 2016;van der Werf et al., 2007).

To examine the effects of drought stress on the onset of autumn leaf senescence, we hypothesized that:

(I)     the timing of the onset of autumn leaf senescence in temperate deciduous trees is advanced by severe drought stress. The leaves of a tree that experiences drought will accumulate the consequences of stress exposure and lose functionality. Therefore, it is likely not beneficial for a tree to maintain active leaves late in the season after severe drought. Instead, to maximize nutrient recovery, trees probably prefer an earlier leaf senescence. In addition, drought would reduce the tree's wood growth and increase its fine root mortality (Brunner et al., 2015;Campioli et al., 2013). Consequently, the tree's carbon sink strength will decline, causing a reduced demand for carbon from the sources (e.g. the leaves) and advance the onset of autumn leaf senescence.

(II)    different tree species show different dynamics in their onset of autumn leaf senescence under drought. We hypothesized that, under drought stress, species with continuous flushing (e.g. birch) will have a more stable timing onset of autumn leaf senescence than species with only one or two leaf flushes during spring-summer (e.g. beech and oak) (Koike, 1990).

We tested these hypotheses by subjecting young trees to treatments comprising less irrigation and warming, and by examining the effect of years with different drought intensities (2017, 2018 and 2019) on mature trees in natural forest stands. Both young and mature trees experienced not only drought, but also heat and increased atmospheric aridity.

# 2. Materials and methods

## 2.1. Study sites and experimental setting

### *2.1.1. Manipulative experiment*

In 2018, we carried out a manipulative experiment at the Drie Eiken Campus in Wilrijk, Belgium (51°09′N, 4°24′E). In early March, 128 individuals of three-year-old beech (*Fagus sylvatica*) saplings, from a local nursery and with the same local provenance, were planted in pots with a volume of 35 liters and a surface area of 0.07 m². The pots were filled with 20% peat and 80% white sand. Eight beech saplings were placed in each of twelve climate-controlled glasshouses with a ground surface of 1.5 x 1.5 m and a height at the north and south side of 1.5 m and 1.2 m, respectively. The glasshouses had a roof of colorless polycarbonate (a 4 mm thick plate) reducing the incoming light by ± 20% and modifying the spectral quality only in the UV range (Kwon et al., 2017). The glasshouses had three sides that could be opened or closed and were equipped with a combined humidity-temperature sensor (QFA66, Siemens, Erlangen, Germany) to monitor the relative humidity and air temperature (Fig. 1, panel A and B) (Kwon et al., 2017). One pot per glasshouse was also equipped with a soil moisture smart sensor (HOBO S-SMD-M005, Onset, MA, USA) to monitor the soil water content (Fig. 1, panel C). The latter sensors became available only at the time the drought stress was alleviated (see below). More details on the set-up of the glasshouses can be found in the literature (Van den Berge et al., 2011;De Boeck et al., 2012;Fu et al., 2014). Two treatments were organized (n = 48 per treatment; see below). In addition to the saplings in the glasshouses, eight beech saplings were placed in each of four reference plots outside of the glasshouses (n = 32, Ref.). The relative humidity and air temperature of the outside reference plots were monitored by a pocket weather meter (Kestrel 3000, Nielsen, PA, USA). Once in April and once in July, all saplings received 35 g of NPK slow-release fertilizer (DCM ECO-XTRA 1) and 1.8 g of micro elements (DCM MICRO-MIX). Using the relative humidity and air temperature data between 7 a.m. and 7 p.m., the vapor pressure deficit was calculated for both treatments (see below) and the reference plots using the formulas of Buck (1981) (Eq. 1; Fig. 1, panel D).

Equation 1

$$e_0 = 613.75 \times \exp((17.502 \times T)/(240.97 + T))$$
$$e = (RH/100) \times e_0$$
$$VPD = e_0 - e$$

where $e_0$ is the saturation vapor pressure (in Pa), T is the temperature (in °C), e is the actual vapor pressure deficit (in Pa), RH is the relative humidity (in %) and VPD is the vapor pressure deficit (in Pa).

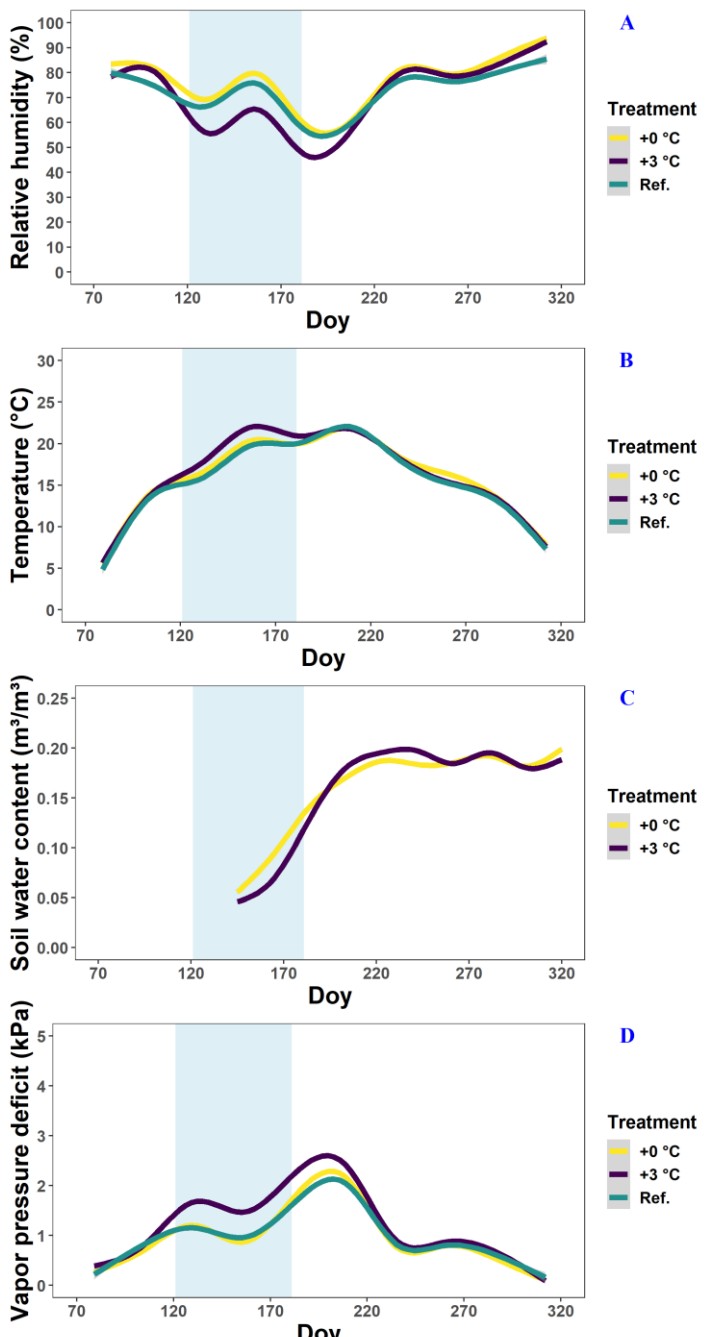

Fig. 1: The relative humidity (panel A), temperature (panel B), soil water content (panel C) and vapor
pressure deficit (panel D) in the glasshouses and outside plots at the Drie Eiken Campus in Wilrijk. Solid
lines represent regressions of half-hourly measurements of the relative humidity (%), temperature (°C),
and soil water content (m³/m³). Regressions were done using generalized additive models implemented
by the *geom_smooth* argument in the R/GGPLOT2 package. The vapor pressure deficit (kPa) was calculated
using the formulas of Buck (1981) using data of the relative humidity and air temperature between 7 a.m.
and 7 p.m. Green, blue and red lines represent the conditions in the reference plots (Ref.), glasshouses
that follow the outside ambient air temperature (+0 °C) and glasshouses that are three degrees warmer
than the outside ambient air temperature (+3 °C), respectively. The light blue band represents the
treatment-period.
kPa).

From planting until April, the saplings were all irrigated two to three times a week until the pots
overflowed. The reference plots outside were maintained with abundant irrigation during the whole
growing season. On the other hand, at the start of the treatment, in early May, we shielded all the
glasshouses using polyethylene film (200 μm thick) and irrigated the saplings only once a week with circa
2.5 liter of water. In addition, we enhanced the drought in six glasshouses by raising the air temperature
three degrees compared to the ambient air temperature (+3 °C). The air temperature in the other six
glasshouses followed the ambient air temperature (+0 °C). There were no significant differences in the
temperature, relative humidity and vapor pressure deficit among the glasshouses with the reference and
+0 °C treatment (Fig. 1). Although no data on the soil water content was available for the reference plots
(due to sensor malfunctioning), we did not expect major drought stress due to their abundant irrigation
and lack of stress signals. Based on this information, the +0 °C treatment can be considered a 'less-
irrigation/drought' treatment. On the other hand, during the treatment, the daily soil water content and
the daily relative humidity in the glasshouses with the +3 °C treatment were significantly lower ($P < 0.001$;
tested using generalized additive mixed models) in comparison to the glasshouses with the +0 °C
treatment. After statistical testing following Rose et al. (2012), the difference between the +0 °C and +3
°C treatments was found to be around 0.025 m³/m³ for the soil water content and 20% for the relative
humidity (Fig. 1; see Data availability). The +3 °C treatment can therefore be considered a combined 'less-
irrigation/drought, warming and increased atmospheric aridity' treatment. In fact, this treatment should
simulate natural drought conditions, which are often associated with heat stress and increased
atmospheric aridity. The plan was to continue the treatment till the end of June but, due to the significant
mortality rate, we were obliged to alleviate the drought already from the 20th of June by increasing the
irrigation to the level of the reference plots. From July, the glasshouses were opened again and the
saplings were irrigated four to five times a week until the end of the season.

A draw-back of the experiment is that the saplings in the reference plots received more incoming light
(i.e. ± 20%) than the saplings in the glasshouses (Van den Berge et al., 2011). However, as beech is a shade
tolerant species, reduced light is unlikely to have limited tree growth. In addition, preliminary tests
suggested that the ratio of light in different wavelengths (e.g. R/FR) during civil twilight (i.e. what is
required for phytochrome to detect the photoperiod) does not change seasonally significantly in our study
area (Chelle et al., 2007).

*2.1.2. Field observations in deciduous forests*
From 2017 to 2019, we monitored dominant mature trees in two forests near Antwerp: the Klein
Schietveld in Kapellen (KS; 51°21'N, 4°37'E) and the Park of Brasschaat (PB; 51°12'N'', 4°26'E). In the KS,
we monitored eight beech trees and eight birch (*Betula pendula*) trees. In the PB, we monitored eight
beech trees and eight oak (*Quercus robur*) trees (thus 32 trees in total). The two forests and their
meteorological conditions are described in detail by Mariën et al. (2019), which also showed a lack of site
effects on the autumn chlorophyll dynamics for the tree species studied here. To have a larger statistical
sample, the data of the two beech stands (also of similar age and stem diameter) were aggregated.

For summer and autumn, we report here the average values for the temperature, precipitation, number
of rainy days, relative humidity, sunshine duration and global solar radiation for the meteorological station
of the Royal Meteorological Institute (KMI) in Ukkel, Belgium (Table 1). For these data, long-term averaged
data was available. The temperature, relative humidity, vapor pressure deficit (see Eq. 1), precipitation
and volumetric soil water content from 2017 to 2019 are presented in more detail using daily values that
were measured at Brasschaat and, whenever necessary, gap-filled with data from the meteorological
station in Woensdrecht, Netherlands (Fig. 2, panel A − B; panel D). The meteorological data from
Brasschaat was provided by the Flemish Institute for Nature and Forest (INBO) and the Integrated Carbon
Observation System (ICOS), while the data from Woensdrecht was provided by the Royal Dutch
Meteorological Institute (KNMI).
Table 1: Overview of the meteorological conditions during the summer and autumn of 2017, 2018 and
2019. All data is measured by the meteorological station of the Royal Meteorological Institute (KMI) in
Ukkel, Belgium (KMI, 2018a, b, 2017b, c, 2019a, b). The degree of abnormality of the values is represented
by two labels: a for abnormal values (with a recurrence time of six years) or e for exceptional values (with
a recurrence time of thirty years). In case only one month had abnormal values, this label is followed by
the name of that particular month. Since 2019, the KMI uses a new system to show the degree of
abnormality: values that are with the five highest values since 1981 are marked by (+), while values within
the three highest values are marked by (++).

| | Normal (1981-2010) | | 2017 | | 2018 | | 2019 | |
|---|---|---|---|---|---|---|---|---|
| | summer | autumn | summer | autumn | summer | autumn | summer | autumn |
| Average temperature (°C) | 17.6 | 10.9 | 18.6 (a) | 11.3 | 19.8 (e) | 11.8 | 19.1 (++) | 11.3 |
| Total precipitation (mm) | 224.6 | 219.9 | 179.9 | 226.5 | 134.7 (a) | 168.5 | 198.6 | 209.3 |
| Average number of rainy days | 43.9 | 51 | 44 | 63 (a) | 20 (e) | 32 (e) | 33 | 53 |
| Relative humidity (%) | 73 | 82 | 67.7 (e, June) | 62 | 62.3 (e, July) | 75 (e, July) | 70 | 83 |
| Sunshine duration (h:m) | 578:20 | 322:00 | 573:21 | 322:00 | 693:06 (a) | 471:12 (e) | 714:38 (++) | 322:23 |
| Global solar radiation (kWh/m²) | 429.6 | 168.2 | 447.1 (a, June) | 233.8 | 498.6 (e, July) | 213.4 (e, October) | 487.9 (+) | 178.4 |


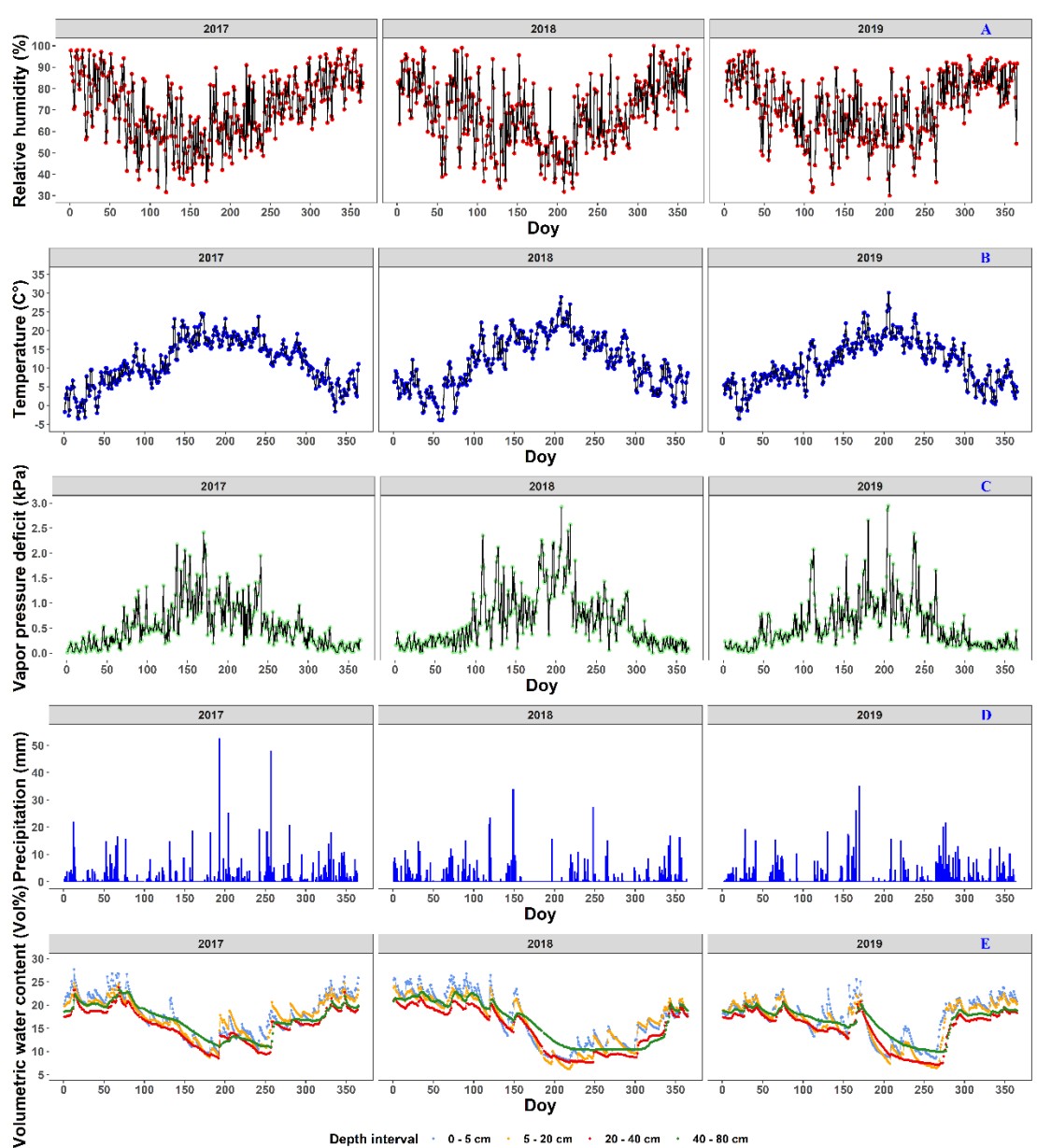

Fig. 2: The meteorological conditions near the Klein Schietveld and Park of Brasschaat. The line plots represent the daily average relative humidity (%; red), temperature (°C; blue) and vapor pressure deficit (kPa; green). The bar plots represent the daily precipitation (mm; light blue). The volumetric soil water content (Vol%) at depth intervals of 0 – 5 cm, 5 – 20 cm, 20 – 40 cm and 40 – 80 cm is presented as line plots in cornflower blue, orange, red and green, respectively. The relative humidity, temperature, vapor pressure deficit and precipitation data was measured every half hour and provided by the Flemish Institute for Nature and Forest (INBO), the Integrated Carbon Observation System (ICOS) and the Royal Dutch Meteorological Institute (KNMI). The vapor pressure deficit (kPa) was calculated using the formulas of Buck (1981) using data of the relative humidity and air temperature between 7 a.m. and 7 p.m. The volumetric soil water content data was first measured every six hours but after 03/07/2018 measurements were made every hour. The volumetric soil water content data was provided through courtesy of INBO.

The distance from Ukkel and Woensdrecht to our sites is 60 km and 20 km, respectively. However, both locations show no major climatological differences with the KS and PB, and are representative for the inter-annual variability experienced by the forests. The station of Ukkel is located within a green area in the suburb of Brussels (thus, classifiable as "urban park"). The microclimate is expected to be different than at our study sites. However, data from Ukkel were used to describe the intra-annual variability and long-term trends in the meteorological variables, which are less affected by the microclimate. The meteorological station of Brasschaat is very close to our sampling site in the Park of Brasschaat and in the Klein Schietveld (± 3 km and ± 4 km, respectively). The meteorological station in Brasschaat is a 40 m high scaffolding tower, at which measurements are taken at various heights, and stands in a patch of mixed forest covered mainly by Scots pines and deciduous tree species, such as oak and birch (see Carrara et al. (2003) for more information). Data of the temperature, precipitation and humidity were taken at the top of the tower. Concurrently, the volumetric soil water content was measured near the scaffolding tower using twelve water reflectometers (CS616 Water Content Reflectometer, Campbell Scientific, UT, USA) connected to a central data logger (CR1000 data logger, Campbell Scientific, UT, USA). The water reflectometers were equally divided over three sampling pits at an 8 m distance from the central data logger. In 2010 and in each pit, the water reflectometers were installed in pedogenetic horizons at four depth intervals (i.e. $0 - 5$ cm, $5 - 20$ cm, $20 - 40$ cm and $40 - 80$ cm). The volumetric soil water content data was first measured every six hours but after 03/07/2018 measurements were made every hour. The volumetric soil water content was calibrated following De Vos (2016) and averaged per day and depth interval. The station of Woensdrecht is located in an open field at a local airport surrounded by heathland and urban area. It is located near the Markiezaatsmeer, an enclosed swamp ecosystem, within the river mouth of the Schelde. The measurements in both Ukkel and Woensdrecht are taken at a height of 1.5 m. However, these data were only used as gap-filling in case of short term gaps in the long-term Brasschaat series.

### 2.1.3. The rainfall deficit: an indicator of drought stress for 2017 - 2019

To indicate the magnitude of the droughts, we computed the rainfall deficit from 2017 to 2019 using data on the relative humidity, solar radiation, wind speed, temperature and precipitation from the meteorological station in Ukkel. Here, the meteorological records go back the longest in Belgium. The rainfall deficit is computed on a daily basis by accumulating the daily potential evapotranspiration minus the daily amount of precipitation. This was done in two ways: (I) per hydrological year, starting from a zero deficit at the start of the hydrological year (1st of April) and (II) continuous computation, so no restart from 0 at the start of each hydrological year. The latter method has the benefit that the long-term effect of accumulated droughts from successive years is accounted for.

The potential evapotranspiration was computed by means of the method of Bultot et al. (1983), which is similar to the method of Penman (1948), but has parameters that are calibrated specifically for the local Belgian conditions. Unlike for the rainfall deficit starting from a zero deficit, we accounted in the calculation of the continuously computed rainfall deficit for the hydrological fraction in wet periods that does not contribute to building up ground water reserves. At the station of Ukkel, daily precipitation and potential evapotranspiration data are available since more than 100 years. The precipitation data are collected since 1898 on the same location, and is measured using the same instrument. For this study, the data for the 100-year period 1901-2000 was considered as the reference period for the computation of long-term statistics on the rainfall deficit.

## 2.2. Measuring autumn leaf senescence: the chlorophyll content index and the loss of canopy greenness

In the manipulative experiment from late-July until late-November, we measured the chlorophyll content index (CCI; a proxy for the chlorophyll concentration) of each tree sapling weekly by randomly selecting one leaf from the outer, middle and inner layer of the upper part of the crown. The CCI was measured using a chlorophyll content meter, which measures the optical absorbance in the 653 nm and 931 nm wavebands (CCM-200 plus, Opti-Sciences Inc., Hudson, NH, USA). Concurrently, we visually estimated the loss of canopy greenness (LOCG; scaled between 0 and 1) of each sapling following the method of Vitasse et al. (2011), which accounts for both the percentage of leaves that have changed color and the percentage of leaves that have fallen.

For half of the monitored mature trees in the two forests and from the end of July to the end of November, tree-climbers collected leaves on eight occasions per year separated by two to three weeks. During each measurement day, they collected five sun-leaves and five shade-leaves from each tree. Afterwards, the CCI was immediately measured on the harvested leaves using the same chlorophyll content meter as described above. From early September to late November, the loss of canopy greenness was estimated in a similar fashion to the manipulative experiment for the 32 mature trees (Vitasse et al., 2011).

Following the method of Mariën et al. (2019), we validated the CCI values by measuring also the chlorophyll concentrations (Fig. A1). In 2017 and 2018, on one occasion per month and using a 10-mm diameter cylinder, we collected samples of leaf tissue from the leaves of the mature trees for which we also measured the CCI. After storage at -80 °C, the samples were grounded using glass beads and a centrifuge. The result was dissolved in ethanol and the absorption of the solution was measured using a spectrophotometer (Smart Spec Plus Spectrophotometer, Bio-Rad) at different wavelengths for Chlorophyll a (662 nm) and chlorophyll b (644 nm). The chlorophyll concentrations could then be derived from the absorption values using the formulas described in Holm (1954) and Vonwettstein (1957).

## 2.3. Tree mortality in the manipulative experiment

In this study, we only considered those trees that defoliated due to autumn leaf senescence. Other tree saplings have died or defoliated completely due to accelerated leaf senescence during or just after the treatment period. Since chlorophyll degradation is a common feature of both senescence processes and nutrient remobilization was only measured indirectly by CCI, we did not consider (I) tree saplings that showed an early or abrupt defoliation (without gradual coloration) before the 18[th] of August (n = 20) and (II) tree saplings with constant CCI values lower than three, the limit at which the values of the CCI meter can be interpreted, for the whole period from August to November (n = 18). Like in other studies, some defoliated tree saplings produced a few new leaves as last attempt to prevent death (Vander Mijnsbrugge et al., 2016;Turcsan et al., 2016). However, there were not enough of such leaves for meaningful analyses.

## 2.4. Statistical analyses

All statistical analyses were performed using R v.3.6.1. (R Core Team, 2020). The model assumptions were tested following Zuur et al. (2010). All graphical output is built using the packages R/GGPLOT2, R/GGPUBR, R/VIRIDIS and R/COWPLOT, while data manipulation has been done using R/DPLYR (Wickham, 2009;Wilke, 2019;Garnier, 2018;Kassambara, 2019;Wickham et al., 2018).

## 2.4.1. Assessing the patterns of CCI and loss of canopy greenness using generalized additive
mixed models
The patterns of the CCI and loss of canopy greenness data from both our tree saplings and mature trees
were assessed using generalized additive mixed models (GAMMs) built using the packages R/MGCV and
R/GRATIA (Wood, 2011;Simpson, 2020;Hastie and Tibshirani, 1986;Pedersen et al., 2019). We used GAMMs
because they allow more flexibility than other models (e.g. generalized linear models) to model the
distribution parameter μ (i.e. the mean of the observed random variable) and the continuous explanatory
variables (Rigby and Stasinopoulos, 2005).

To model the CCI of both our tree saplings and mature trees as a function of their covariates, Gaussian
GAMMs with the identity link function were used (Table 2). To model the loss of canopy greenness of both
our tree saplings and mature trees as a function of their covariates and because the loss of canopy
greenness is scaled between 0 and 1, Binomial GAMMs with the logistic link function were used (Table 2).
The GAMMs were chosen with the lowest AIC value (Akaike information criterion) and all factor-smooth
interaction terms were smoothed using P-splines to address the large gap in data (i.e. from November to
June) between the yearly sampling periods.

For the CCI of the beech saplings, the fixed covariates were the *treatment* (categorical with three levels),
*leaf place* (categorical with three levels) and *day of the year* (continuous; model 1). The interaction term
was modelled as a factor-smooth interaction between the covariates *day of the year* and *treatment*. The
dependency among observations of the same individual tree was incorporated by using *individual tree* as
random intercept.
Model 1
$$Y_{ij} \sim \text{Gaussian}(\mu_{ij}, \text{cst.})$$
$$g(\mathbb{E}(Y_{ij})) = g(\mu_{ij})$$
$$g(\mu_{ij}) = \text{Treatment}_{ij} + \text{Leaf place}_{ij} + f(\text{Day of the year}_{ij}, \text{Treatment}_{ij}) + \text{Individual tree}_i$$

where g is the identity link function, $\mu_{ij}$ is the conditional mean, $Y_{ij}$ is the *j*th observation of the response
variable (i.e. the CCI) in Individual tree *i*, and *i* = 1,…, 128, and Individual tree$_i$ is the random intercept (Zuur
et al., 2007;Zuur et al., 2016).
For the loss of canopy greenness of the beech saplings, the fixed covariates were the *treatment*
(categorical with three levels) and *day of the year* (continuous; model 2). The interaction term and the
dependency among observations of the same individual tree were treated as in model 1.
Model 2
$$Y_{ij} \sim B(n_{ij}, \pi_{ij})$$
$$g(\mathbb{E}(Y_{ij})) = g(\mu_{ij})$$
$$g(\mu_{ij}) = \text{Treatment}_{ij} + f(\text{Day of the year}_{ij}, \text{Treatment}_{ij}) + \text{Individual tree}_i$$

where $n_{ij}$ is the number of observations, $\pi_{ij}$ is the probability of 'success', g is the logit link function, $\mu_{ij}$ is
the conditional mean, $Y_{ij}$ is the *j*th observation of the response variable (i.e. the loss of canopy greenness)
in Individual tree *i*, and *i* = 1,…, 128, and Individual tree$_i$ is the random intercept.
For the CCI of the mature beech, birch and oak trees, the fixed covariates were the *year* (categorical with
three levels), *leaf type* (categorical with two levels) and *day of the year* (continuous; model 3). The
interaction term was modelled as a factor-smooth interaction between the covariates *day of the*
*year* and *Year*. The dependency among observations of the same individual tree was incorporated
using *individual tree* as random intercept.
Model 3
$$Y_{ij} \sim \text{Gaussian}(\mu_{ij}, \text{cst.})$$
$$g(\mathbb{E}(Y_{ij})) = g(\mu_{ij})$$
$$g(\mu_{ij}) = \text{Year}_{ij} + \text{Leaf type}_{ij} + f(\text{Day of the year}_{ij}, \text{Year}_{ij}) + \text{Individual tree}_i$$

where g is the identity link function, $\mu_{ij}$ is the conditional mean, $Y_{ij}$ is the *j*th observation of the response
variable (i.e. the CCI) in Individual tree *i*, and *i* = 1,…, 8 for beech, *i* = 1,…, 4 for birch and *i* = 1,…, 4 for oak,
and Individual tree$_i$ is the random intercept.
For the loss of canopy greenness of the mature beech, birch and oak trees, the fixed covariates were the
*Year* (categorical with three levels) and *day of the year* (continuous; model 4). The interaction term and
the dependency among observations of the same individual tree were treated as in model 3.
Model 4
$$Y_{ij} \sim B(n_{ij}, \pi_{ij})$$
$$g(\mathbb{E}(Y_{ij})) = g(\mu_{ij})$$
$$g(\mu_{ij}) = \text{Year}_{ij} + f(\text{Day of the year}_{ij}, \text{Year}_{ij}) + \text{Individual tree}_i$$

where $n_{ij}$ is the number of observations, $\pi_{ij}$ is the probability of 'success', g is the logit link function, $\mu_{ij}$ is
the conditional mean, $Y_{ij}$ is the *j*th observation of the response variable (i.e. the loss of canopy greenness)
in Individual tree *i*, and *i* = 1,…, 16 for beech, *i* = 1,…, 8 for birch and *i* = 1,…, 8 for oak, and Individual
tree$_i$ is the random intercept.

Table 2: Adjusted R², effective degrees of freedom (edf) and F-test values of the GAMM smooth terms (*Day of the year*). All smooth terms were significant, with p-values < 0.001. $\mathbb{E}(y_i)$ are the expected values of the response variable $y_i$, $f(x_i)$ is the smooth function of the covariate $x_i$, $\beta_i$ is the intercept of the covariate $x_i$, $\zeta$ is the random effect and $\varepsilon_i$ are the errors. All smooth functions were fitted using P-splines. The chlorophyll content index, loss of canopy greenness, day of the year and tree individual are abbreviated by CCI, LOCG, Doy and ID, respectively.

| Site | Species | $Y_i$ | Model equation | Family distribution | Link function | AIC | Adjusted R² | Smooth term | Treatment | Edf | F or Chi.sq |
|---|---|---|---|---|---|---|---|---|---|---|---|
| Wilrijk | *Fagus sylvatica* | CCI | (1) $g(\mathbb{E}(y_i)) = f_1 Treatment_i(Doy_i) + \beta_1 Treatment_i + \beta_2 Leaf\_place_i + \zeta_{ID} + \varepsilon_i$ | Gaussian | Identity | 17373 | 0.61 | Day of the year | Reference | 4.8 | 337.5 |
| | | | | | | | | | +0 °C | 5.8 | 175 |
| | | | | | | | | | +3 °C | 6.1 | 34.4 |
| Wilrijk | *Fagus sylvatica* | Loss of canopy greenness | (2) $g(\mathbb{E}(y_i)) = f_1 Treatment_i(Doy_i) + \beta_1 Treatment_i + \zeta_{ID} + \varepsilon_i$ | Binomial | Logit | 878 | 0.76 | Day of the year | Reference | 3.6 | 112.6 |
| | | | | | | | | | +0 °C | 1.1 | 105.9 |
| | | | | | | | | | +3 °C | 1 | 53.7 |
| | | | | | | | | | **Year** | | |
| KS & PB | *Fagus sylvatica* | CCI | (3) $g(\mathbb{E}(y_i)) = f_1 Year_i(Doy_i) + \beta_1 Year_i + \beta_2 Leaf\_type_i + \zeta_{ID} + \varepsilon_i$ | Gaussian | Identity | 9382 | 0.7 | Day of the year | 2017 | 4.6 | 197.8 |
| | | | | | | | | | 2018 | 5.3 | 221.6 |
| | | | | | | | | | 2019 | 5.2 | 193.2 |
| KS & PB | *Fagus sylvatica* | Loss of canopy greenness | (4) $g(\mathbb{E}(y_i)) = f_1 Year_i(Doy_i) + \beta_1 Year_i + \zeta_{ID} + \varepsilon_i$ | Binomial | Logit | 450 | 0.87 | Day of the year | 2017 | 2.4 | 44.8 |
| | | | | | | | | | 2018 | 2.5 | 70.6 |
| | | | | | | | | | 2019 | 2.7 | 66 |
| KS | *Betula pendula* | CCI | (5) $g(\mathbb{E}(y_i)) = f_1 Year_i(Doy_i) + \beta_1 Year_i + \beta_2 Leaf\_type_i + \zeta_{ID} + \varepsilon_i$ | Gaussian | Identity | 4546 | 0.44 | Day of the year | 2017 | 3.2 | 25.9 |
| | | | | | | | | | 2018 | 5 | 56.9 |
| | | | | | | | | | 2019 | 3.1 | 14.7 |
| KS | *Betula pendula* | Loss of canopy greenness | (6) $g(\mathbb{E}(y_i)) = f_1 Year_i(Doy_i) + \beta_1 Year_i + \zeta_{ID} + \varepsilon_i$ | Binomial | Logit | 254 | 0.89 | Day of the year | 2017 | 1 | 20.6 |
| | | | | | | | | | 2018 | 1 | 36 |
| | | | | | | | | | 2019 | 1.6 | 48.2 |
| PB | *Quercus robur* | CCI | (7) $g(\mathbb{E}(y_i)) = f_1 Year_i(Doy_i) + \beta_1 Year_i + \beta_2 Leaf\_type_i + \zeta_{ID} + \varepsilon_i$ | Gaussian | Identity | 5694 | 0.52 | Day of the year | 2017 | 3.3 | 62.5 |
| | | | | | | | | | 2018 | 5.1 | 84.4 |
| | | | | | | | | | 2019 | 4.3 | 30.7 |
| PB | *Quercus robur* | Loss of canopy greenness | (8) $g(\mathbb{E}(y_i)) = f_1 Year_i(Doy_i) + \beta_1 Year_i + \zeta_{ID} + \varepsilon_i$ | Binomial | Logit | 225 | 0.85 | Day of the year | 2017 | 1.2 | 12.5 |
| | | | | | | | | | 2018 | 1.9 | 33.6 |
| | | | | | | | | | 2019 | 2.4 | 32 |

### 2.4.2. Using breakpoints to indicate the onset of autumn leaf senescence and the onset of the loss of canopy greenness

In principle, the onset of autumn leaf senescence could be derived from the CCI or loss of canopy greenness. However, Mariën et al. (2019) recently showed that the latter method cannot be used under severe drought stress. Therefore, two phenological variables were considered to describe the autumn canopy dynamics: the onset of autumn leaf senescence derived from the CCI (the onset of autumn leaf senescence) and the onset of the loss of canopy greenness. For each tree, we defined the onset of autumn leaf senescence and the onset of loss of canopy greenness as the date by which the variable of interest started to decline substantially in early autumn. These dates were calculated using piecewise linear regressions and are represented by the breakpoints resulting from these analyses (Menzel et al., 2015;Mariën et al., 2019;Xie and Wilson, 2020). The piecewise linear regressions were performed using R/SEGMENTED (Vito and Muggeo, 2008). The uncertainty reported represents the inter-tree variability. Trees that did not show a clear breakpoint (13 in the manipulative experiment) were not considered in the analysis. These trees did not show a different pattern of CCI or loss of canopy greenness than the other trees (Fig. A2).

### 2.4.3. Comparing the onset of autumn leaf senescence among tree saplings exposed to different treatments

We tested whether the beech saplings exposed to the three treatments in 2018 differed in their onset of autumn leaf senescence using a linear model with the onset of autumn leaf senescence as response variable and *treatment* (categorical with three levels) as fixed covariate. The residuals of the model were approximately normally distributed and a Breusch-Pagan test, the R/ncvTest and R/bptest in the R/CAR and R/LMTEST packages, showed no evidence of heteroscedasticity ($P > 0.05$) (Fox and Weisberg, 2019;Zeileis and Hothorn, 2002). A one-way ANOVA was used to detect significant differences in the onset of autumn leaf senescence among the treatments.

### 2.4.4. Comparing the onset of autumn leaf senescence and the onset of loss of canopy greenness in mature trees among species and years

To model the onset of autumn leaf senescence and the onset of the loss of canopy greenness as a function of their covariates, Gaussian linear mixed models were used. These models were built with the package R/LME4 (Bates et al., 2015).

The effect of the year on the onset of autumn leaf senescence and the onset of the loss of canopy greenness was assessed using two linear mixed effect models with the onset of autumn leaf senescence and the onset of the loss of canopy greenness from the mature beech, birch and oak trees as response variable. The fixed covariate in these two models was the *Year* (categorical with three levels; model 5). To incorporate the dependency among observations of the same species, we used *species* as random intercept.

Model 5

$$Y_{ij} \sim Gaussian(\mu_{ij}, cst.)$$
$$g(\mathbb{E}(Y_{ij})) = g(\mu_{ij})$$
$$\mu_{ij} = Year_{ij} + Species_i$$

where g is the identity link function, $\mu_{ij}$ is the conditional mean, $Y_{ij}$ is the $j$th observation of the response variable in Species $i$, and $i = 1,…, 3$ and $Species_i$ is the random intercept.

The effect of the species on the onset of autumn leaf senescence and the onset of the loss of canopy greenness was assessed using two linear mixed effect models with the onset of autumn leaf senescence and the onset of the loss of canopy greenness from the mature beech, birch and oak trees as response variable. The fixed covariate in these two models was the *Species* (categorical with three levels; model 6). To incorporate the dependency among observations of the same year, we used *Year* as random intercept.

Model 6

$$Y_{ij} \sim \text{Gaussian}(\mu_{ij}, \text{cst.})$$
$$g(\mathbb{E}(Y_{ij})) = g(\mu_{ij})$$
$$\mu_{ij} = \text{Species}_{ij} + \text{Year}_i$$

where g is the identity link function, $\mu_{ij}$ is the conditional mean, $Y_{ij}$ is the *j*th observation of the response variable in Year *i*, and *i* = 1,…, 3 and Year$_i$ is the random intercept.

The residuals of the models were approximately normally distributed and showed no heteroscedasticity (tested using diagnostic plots). Therefore, we used Pearson's chi-square test, R/drop1 in the R/LME4 package, to detect significant differences in the onset of autumn leaf senescence and the onset of the loss of canopy greenness among the predictor variables. A multiple comparison test, the R/glht test with method Tukey in the R/MULTCOMP package, was used to test for significant differences among the means of the levels in the predictor variables (Hothorn et al., 2008).

# 3. Results

## 3.1. Magnitude of the drought stress in 2017, 2018 and 2019

The weather in 2018 and 2019 was exceptional, as can be seen in the overview of the meteorological conditions from 2017 to 2019 against the long-term reference values in Table 1 and Figure 2. In 2017, the weather during spring was dry and warm but the weather during summer and autumn was relatively normal (KMI, 2017b, c, a). In contrast, the warm and dry summer of 2018 was marked by abnormal (with an average return time of 6 years) to exceptional (with an average return time of 30 years or more) values (KMI, 2018b). Furthermore, the autumn of 2018 was abnormally dry and all precipitation fell on relatively few days (32) (KMI, 2018a). In the summer of 2019, the average air temperature and the total amount of sunshine were both among the three highest values recorded since 1981. In fact, the absolute maximum air temperature record for Belgium was broken in 2019 (KMI, 2019b). On the other hand, the autumn of 2019 was considered normal (KMI, 2019a).

The rainfall deficit for each day in the hydrological year (from the 1$^{st}$ of April until the 31$^{st}$ of March) and different return times are shown in Figure 3 (panel A & B). This demonstrates that in the late spring of 2017, the summer of 2018 and the summer of 2019 the rainfall deficit reached a return time between 20 and 50 years, 50 years, and 20 years, respectively. The hydrological summers of 2017, 2018 and 2019 had therefore moderate to extremely dry conditions, which led to accumulated rainfall deficit conditions over time (see Figure 3; panel A). Especially the hydrological year starting in 2018 ended with a strong rainfall deficit of about 150 mm, which was not reduced during 2019. The effects of this strong rainfall deficit are also apparent in the lower volumetric soil water content values (ca. 5% less) measured at the beginning of 2019, compared to the same measurements in 2017 and 2018.

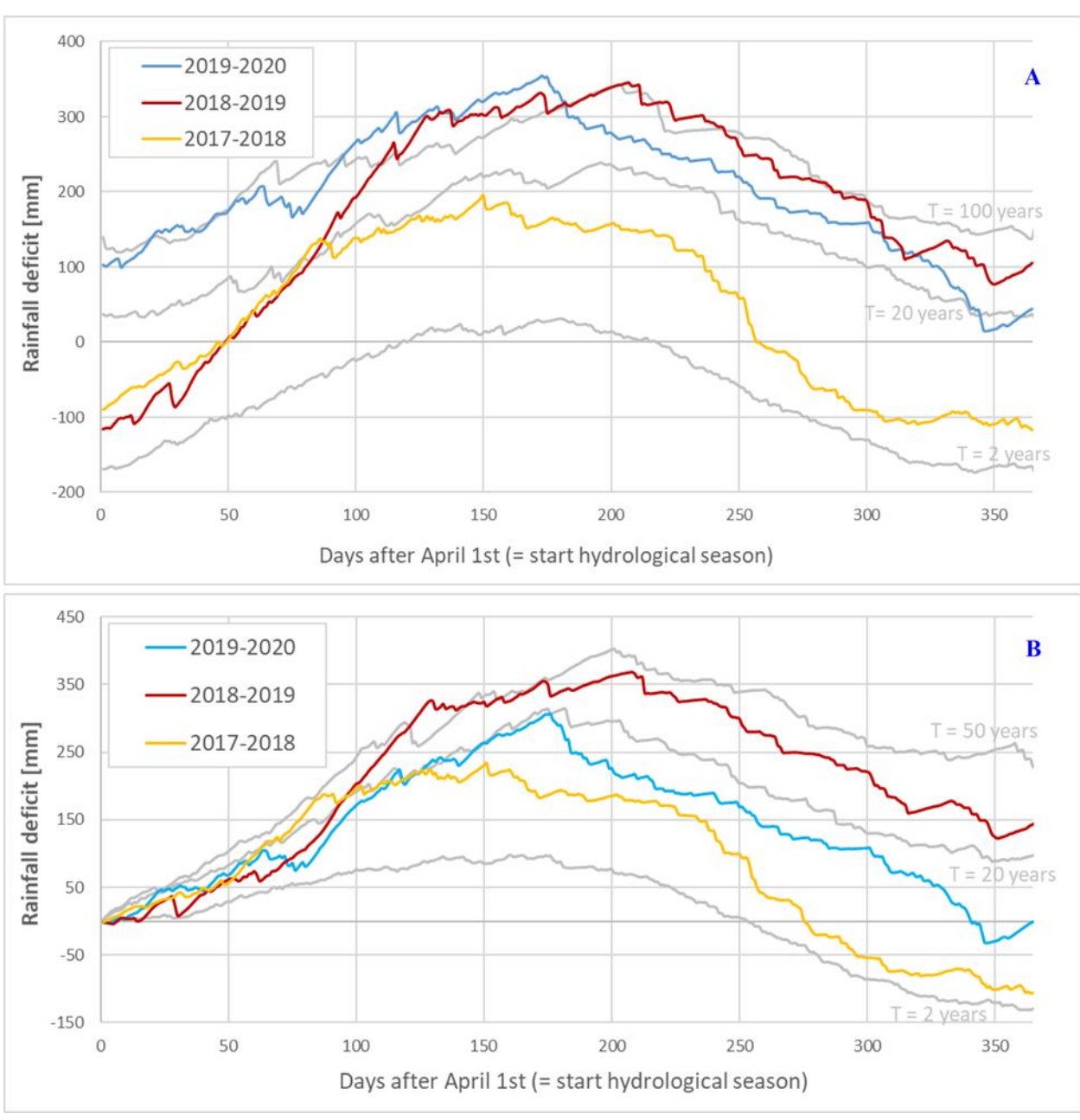

487

Fig. 3: The rainfall deficit for the meteorological station of the Royal Meteorological Institute (KMI) in
Ukkel, Belgium. The colored solid lines represent the rainfall deficit for the hydrological years in the period
2017-2020, while the grey solid lines represent the long-term reference statistics (computed for the 100-
year period 1901 - 2000) with T as the return period, which represents the mean time between two
successive exceedances of a given deficit value and is computed in an empirical way (Willems, 2000, 2013).
Panel A uses a continuous computation, while panel B starts from a zero deficit on the first of April (the
start of the hydrological year). The colors represent the rainfall deficit in 2017 (light blue), 2018 (red) and
2019 (yellow).

496

## 3.2. The effect of drought, heat stress and increased atmospheric aridity on the onset of autumn leaf senescence in tree saplings in the manipulative experiment

For all treatments, the CCI values of the beech saplings showed an overall moderate decrease until the beginning of October. Afterwards, this decrease accelerated (Fig. 4; panel A & C; Table 2). In the +0 °C and especially the +3 °C treatment, an abnormal CCI decline was observed in early August with only a partial recovery later on. As a result, from the beginning of August until mid-September, the CCI values of the beech saplings in the reference plots were significantly higher than the CCI values of the beech saplings in the glasshouses. From the end of September, the CCI decreased in all treatments, showing similar CCI measurements across treatments. However, the modeled CCI of the +3 °C treatment declined slower than the modeled CCI of the other two treatments. No significant difference was detected in the timing of the onset of autumn leaf senescence among the beech saplings exposed to the three different treatments, as the mean onset of autumn leaf senescence was between the 21$^{st}$ (DOY = 260 ± 5) and 25$^{th}$ (DOY = 264 ± 4) of September ($P$ = 0.7; Fig. A3).

The canopy greenness for the beech saplings showed a stable decline from early August until the end of autumn (Fig. 4; panel B & D; Table 2). Nevertheless, during September, the canopy greenness of the beech saplings in the reference plots was significantly higher than the canopy greenness of the beech saplings in the glasshouses with the +3 °C treatment.

The tree saplings in the glasshouses of both treatments were exposed to a high mortality with 14% and 26% of the tree saplings in the glasshouses with the +0 °C and +3 °C treatment, respectively, considered 'dead' along our criteria (see §2.3.). In the reference plots, no beech saplings died.

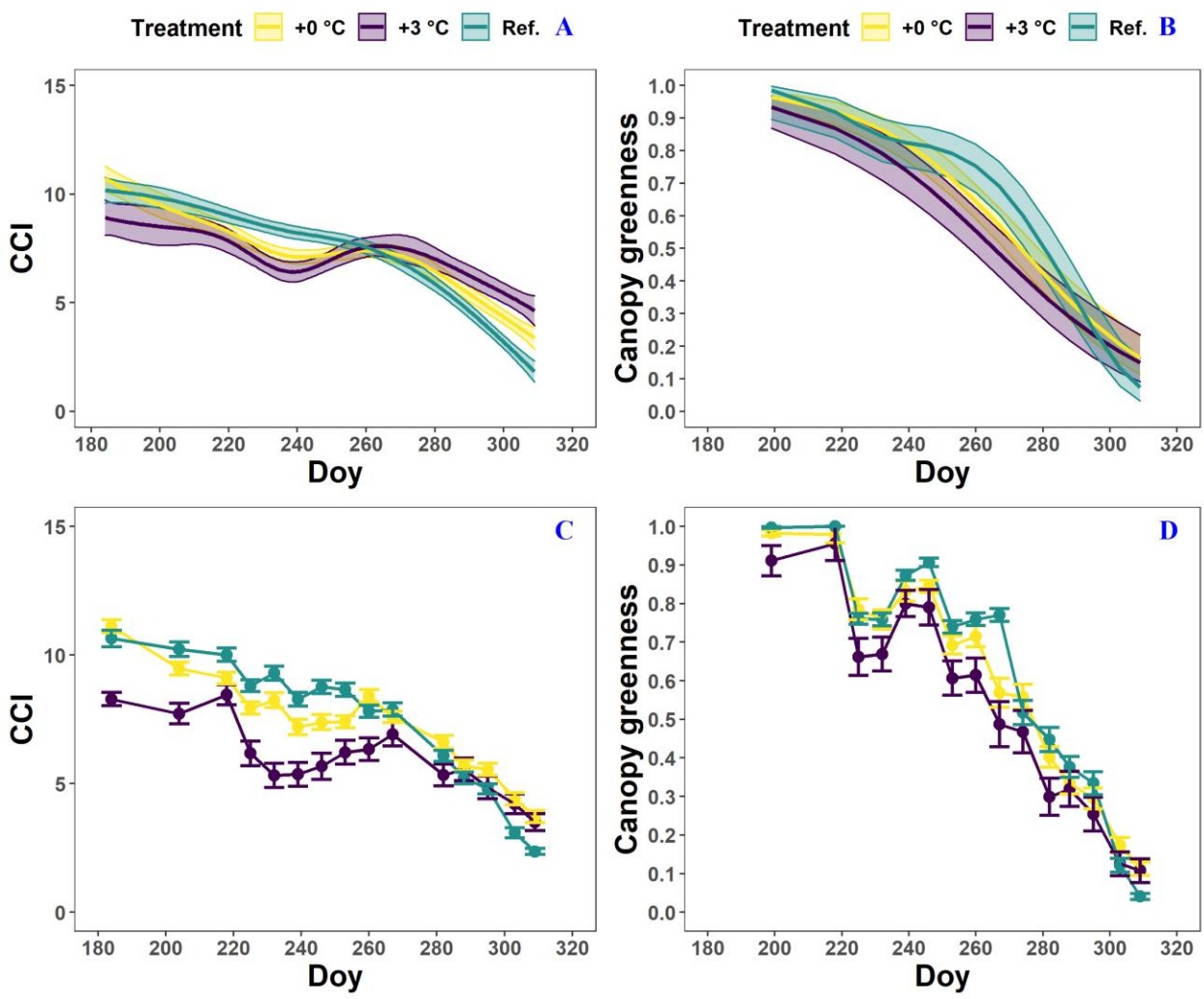

519

Fig. 4: The generalized additive mixed model fits for the chlorophyll content index (CCI; panel A) and loss
of canopy greenness (panel B) of the *Fagus sylvatica* saplings at the Drie Eiken Campus in Wilrijk. The
colored solid lines represent smooth terms, while the colored shaded bands around the smooth terms
approximate the 95% simultaneous confidence intervals (panel A) and 95% pointwise confidence intervals
(panel B). The dots and error bars represent the mean CCI (panel C) and mean canopy greenness (panel
D) with standard errors. The colors represent the CCI or the loss of canopy greenness of the beech saplings
in the reference plots (green; Ref.), the glasshouses that followed the outside ambient air temperature
(yellow; +0 °C) and the glasshouses that were three degrees warmer than the outside ambient air
temperature (purple; +3 °C), respectively.

529

## 3.3. Inter-annual and inter-species variability in the timing of the onset of autumn leaf senescence and the onset of the loss of canopy greenness in mature trees

The pattern in the CCI values for the mature beech, birch and oak trees seems consistent throughout the years with stable values in summer and a rapid decline around late October (Fig. 5 - 7; panel A & C; Table 2). We also observed no significant difference in the onset of autumn leaf senescence among the years ($P$ = 0.09) and species ($P$ = 1). The mean onset of autumn leaf senescence among the years was from the 8th (DOY = 281 ± 6) to the 19th (DOY = 292 ± 6) of October (Fig. A4; panel A), while the mean onset of autumn leaf senescence among the species was around the 13th of October (DOY = 286 ± 6; Fig. A4; panel B). The CCI correlated linearly with the chlorophyll concentrations but the data showed more variation in 2018 than 2017 (see Fig. A1).

The pattern in the canopy greenness for the mature beech, birch and oak trees seemed less consistent throughout the years (Fig. 5 - 7; panel B & D; Table 2). The loss of canopy greenness showed a very similar pattern between 2017 and 2019 for birch and beech, with the start of the decline in canopy greenness values around late September for birch and late October for beech. Like beech and birch, oak showed a standard pattern in 2019 with the start of the seasonal decline in late October. However, in 2017, oak showed an earlier loss of canopy greenness with the start of the seasonal decline in mid-September. In all cases, a rapid decline in the canopy greenness was observed in late autumn. In 2018, all species showed an earlier and steeper decline in their canopy greenness values. This effect was also reflected by a significant difference in the onset of the loss of canopy greenness among the years ($P$ = 5 x $10^{-11}$). Across species, the onset of the loss of canopy greenness did not differ significantly ($P$ = 0.9) between 2017 (DOY = 292 ± 9) and 2019 (DOY = 290 ± 4), while it occurred 26 and 25 days earlier in 2018 (DOY = 266 ± 4) compared to 2017 ($P$ = 1 x $10^{-5}$) and 2019 (P = 1 x $10^{-5}$), respectively (Fig. A5; panel A). However, all tree species differed significantly in their onset of the loss of canopy greenness across years ($P$ = 6 x $10^{-9}$). Compared to birch (DOY = 268 ± 9; Fig. A5; panel B), the onset of the loss of canopy greenness for beech was on average 16 days later ($P$ = 1 x $10^{-4}$; DOY = 284 ± 4), while for oak this was 30 days later ($P$ = 1 x $10^{-4}$; DOY = 298 ± 4). The onset of the loss of canopy greenness for beech was also 14 days earlier than that for oak ($P$ = 7 x $10^{-4}$).

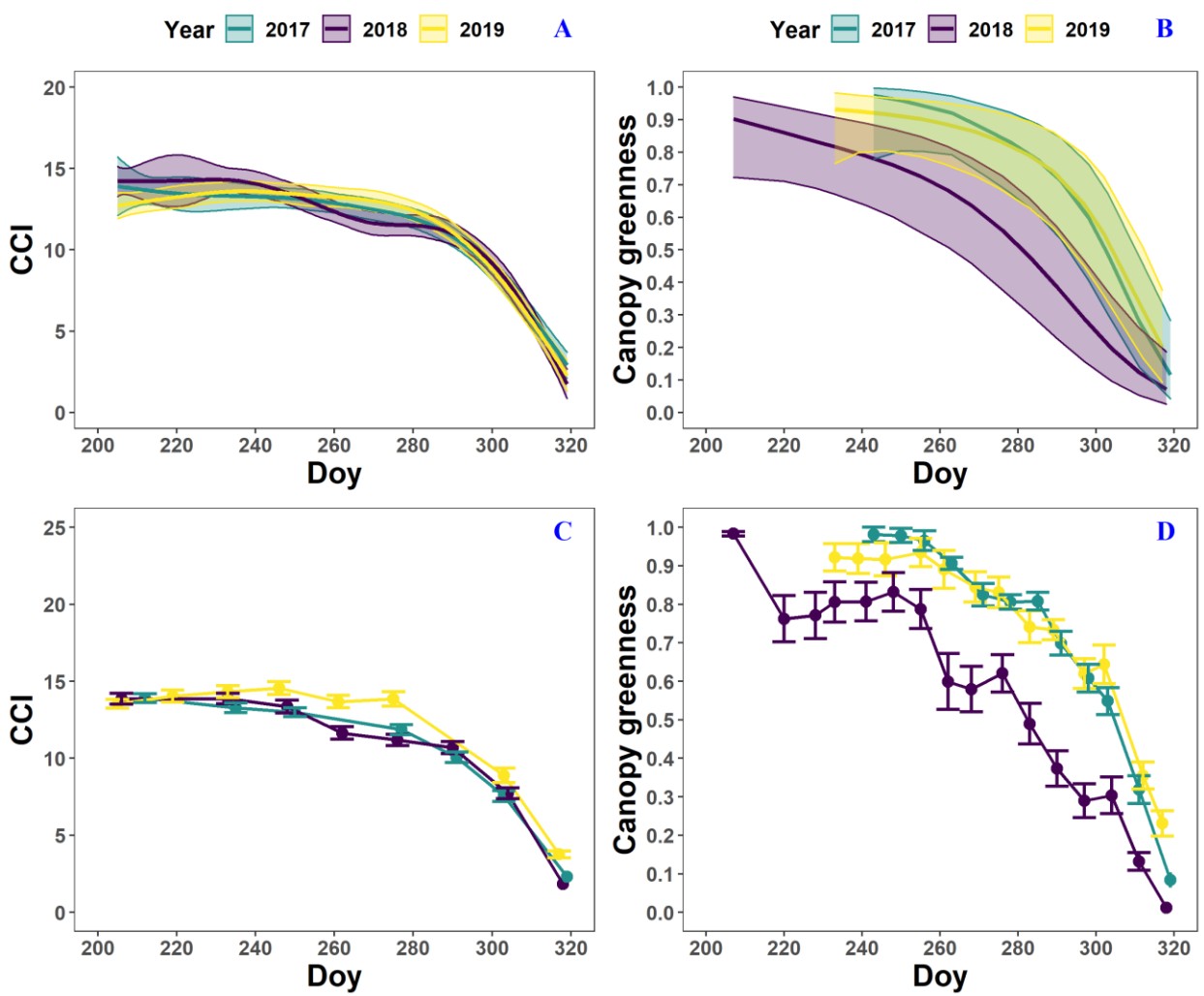

559

Fig. 5: The generalized additive mixed model fits for the chlorophyll content index (CCI; n = 8; panel A) and
loss of canopy greenness (n = 16; panel B) of the mature *Fagus sylvatica* trees at the Klein Schietveld and
Park of Brasschaat. The colored solid lines represent smooth terms, while the colored shaded bands
around the smooth terms represent approximate 95% simultaneous confidence intervals (panel A) and
95% pointwise confidence intervals (panel B). The dots and error bars represent the mean CCI (panel C)
and mean canopy greenness (panel D) with standard errors. The colors represent the CCI or the loss of
canopy greenness of the mature beech trees in 2017 (green), 2018 (purple) and 2019 (yellow).


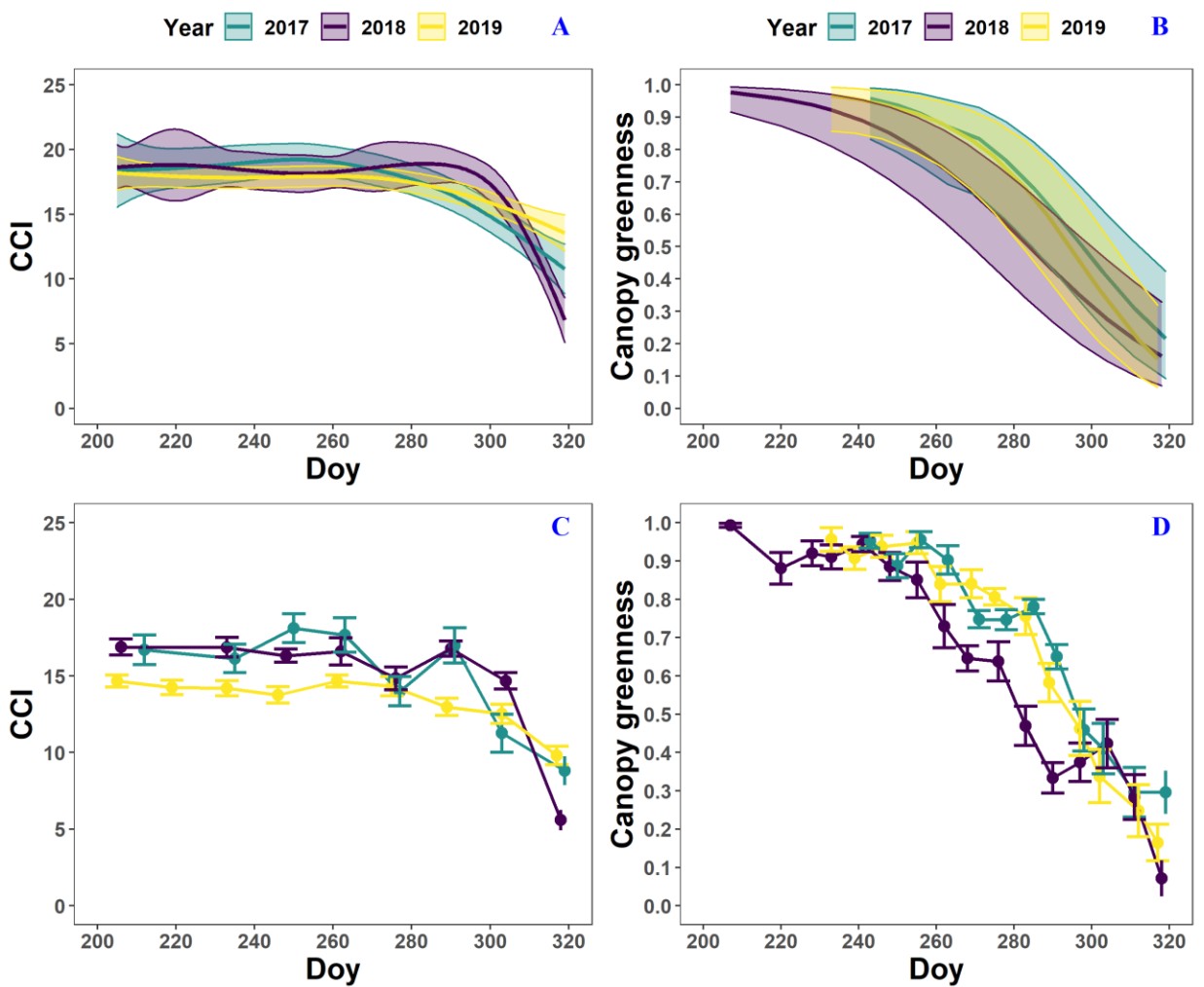

Fig. 6: The generalized additive mixed model fits for the chlorophyll content index (CCI; n = 4; panel A) and
loss of canopy greenness (n = 8; panel B) of the mature *Betula pendula* trees at the Klein Schietveld. The
colored solid lines represent smooth terms, while the colored shaded bands around the smooth terms
represent approximate 95% simultaneous confidence intervals (panel A) and 95 % pointwise confidence
intervals (panel B). The dots and error bars represent the mean CCI (panel C) and mean canopy greenness
(panel D) with standard errors. The colors represent the CCI or the loss of canopy greenness of the mature
birch trees in 2017 (green), 2018 (purple) and 2019 (yellow).

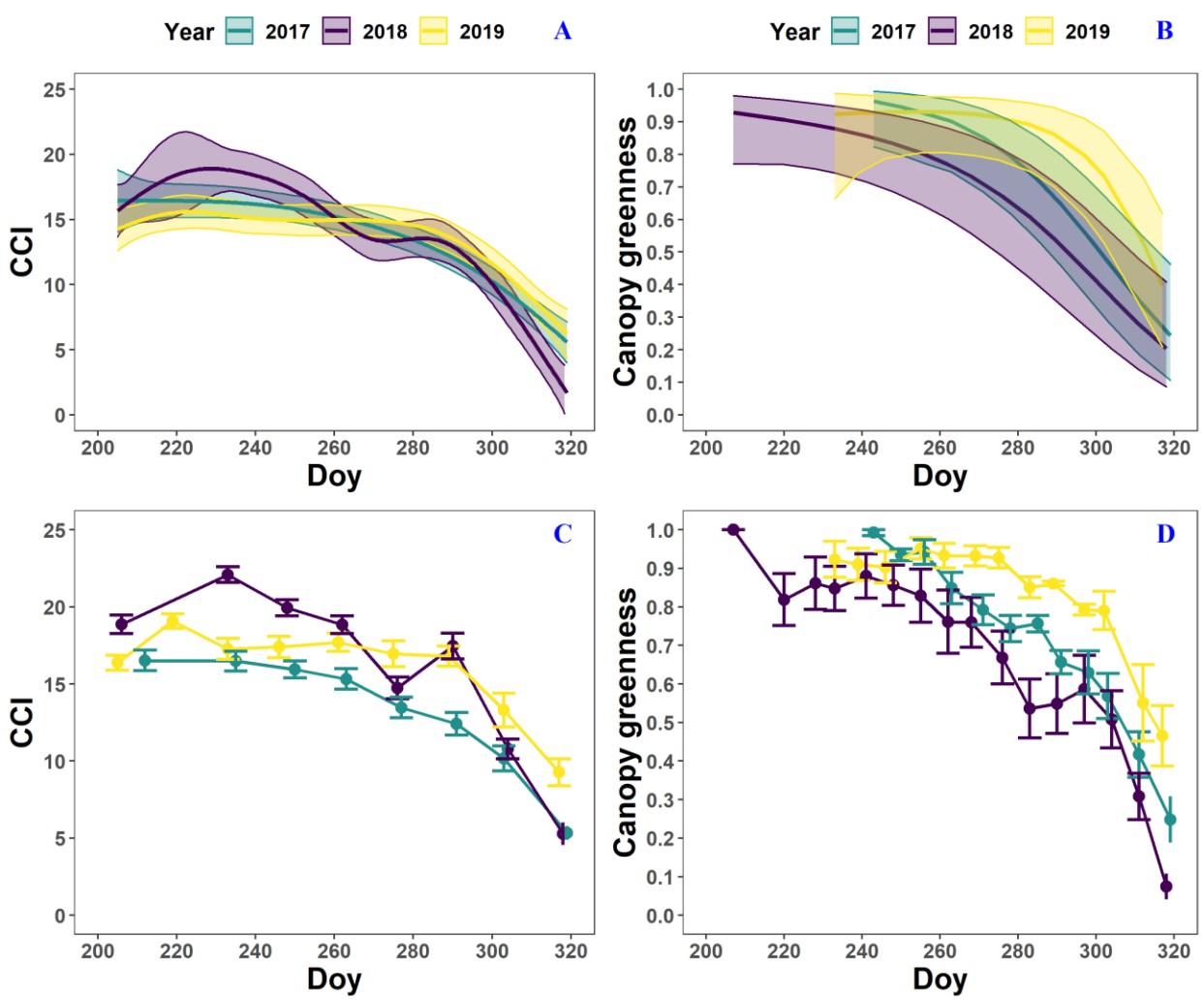

Fig. 7: The generalized additive mixed model fits for the chlorophyll content index (CCI; n = 4; panel A) and
loss of canopy greenness (n = 8; panel B) of the mature *Quercus robur* trees at the Park of Brasschaat. The
colored solid lines represent smooth terms, while the colored shaded bands around the smooth terms
represent approximate 95% simultaneous confidence intervals (panel A) and 95% pointwise confidence
intervals (panel B). The dots and error bars represent the mean CCI (panel C) and mean canopy greenness
(panel D) with standard errors. The colors represent the CCI or the loss of canopy greenness of the mature
oak trees in 2017 (green), 2018 (purple) and 2019 (yellow).

## 4. Discussion

Our results showed that the timing of the onset of autumn leaf senescence in both tree saplings and mature trees was not significantly altered by severe drought, heat stress and increased atmospheric aridity induced by a decline in the soil moisture, relative humidity, and an increase in the air temperature and vapor pressure deficit. These results are in contrast to other studies reporting, for example, that drought stress delays the onset of autumn leaf senescence (determined using remote sensing indices or visual assessment) (Wang et al., 2016;Vander Mijnsbrugge et al., 2016;Zeng et al., 2011;Gárate-Escamilla et al., 2020;Seyednasrollah et al., 2020). However, in our study, drought, heat stress and increased atmospheric aridity did affect the loss of CCI and canopy greenness of our beech saplings, their mortality, and the onset of the loss of canopy greenness in our mature trees. The effect of the drought, heat stress and increased atmospheric aridity on the loss of canopy greenness might be due to an early leaf abscission in response to hydraulic failure of the branches (Wolfe et al., 2016;Munné-Bosch and Alegre, 2004). The manipulation experiment on the beech saplings also revealed that the 'drought/less irrigation' treatment alone (the + 0°C treatment) had less impact (e.g. lower tree mortality, lower premature degradation of chlorophyll in summer) than the combined 'drought/less irrigation, warming and increased atmospheric aridity' treatment (the + 3°C treatment). The decline in the CCI of the saplings exposed to the +3°C treatment, around mid-August, might indicate that physiological damage due to stress can accumulate and become apparent even though stress is alleviated.

Our experimental design did not allow disentangling the effect of the three different stressors within the + 3°C treatment (i.e. drought/less irrigation, warming and increased atmospheric aridity). However, Fu et al. (2018) found that summer warming delayed senescence in beech. In addition, Kint et al. (2012) found that growth in beech is primarily controlled by the water deficit and low relative humidity values during summer. Therefore, the effects observed in the + 3°C treatment might be mainly related to the atmospheric aridity. For the mature trees, the different drought response of the autumn pattern of chlorophyll (no effect) and the loss of canopy greenness (advanced and enhanced) is probably an important reason of confusion still present today in the literature on the relationship between drought and autumn senescence. While the detoxification of chlorophyll is a prerequisite for the expression of different coloration values, chlorophyll does not degrade at the same speed as other leaf pigments. In fact, not even all leaf pigments degrade (or are formed) at the same velocity throughout the senescence process (Keskitalo et al., 2005). Consequently, observations of changing coloration levels are difficult to interpret. Moreover, note that coloration measurements also take into account leaf yellowing and mortality due to hydraulic failure.

The continuously computed rainfall deficit was similar between 2018 and 2019. Nevertheless, the loss of canopy greenness suggests that the drought of 2019, which coincided with several heat waves and increased atmospheric aridity, might have been less damaging for the late-summer leaf dynamics than the drought of 2018 (which lasted longer). The rainfall deficit starting from a zero deficit supports the observation that, despite the accumulated drought effect, the drought of 2019 was less severe in the growing season than the drought of 2018. Perhaps, the conditions of 2018 (i.e. sunny and warm with high vapor pressure deficits, and a long period with a low soil moisture starting earlier than in 2019) triggered the damaging process of cavitation in the trees, while this might have occurred less intensively in 2019 if the stomatal conductance was lower (Barigah et al., 2013;Bolte et al., 2016;Banks et al., 2019). Alternatively, the difference in the timing of the drought peaks (i.e. the drought of 2018 peaked around one month and half earlier than the drought of 2019, Fig. 3A) could have led to divergent responses due to differences in drought sensitivity along the growing season (Banks et al., 2019).

The drought (but also the heat stress and increased atmospheric aridity) did not affect the onset of
autumn leaf senescence of both the beech saplings and the mature trees. Deciduous trees therefore seem
to have a conservative strategy concerning the timing of their autumn leaf senescence that might be under
the control of a constant variable (e.g. the day-length or spectral quality) (Michelson et al., 2018;Chiang
et al., 2019). Such a strategy prioritizes carbon uptake over nutrient remobilization, as a fixed onset of
autumn leaf senescence would not allow an advanced nutrient remobilization when required (Keskitalo
et al., 2005;Brelsford et al., 2019). Moreover, such a strategy makes the trees vulnerable against the
effects of early frost. In case of early frost, the trees might not complete their nutrient resorption. Possible
consequences of an incomplete nutrient resorption over a longer time period might include a decline in
the overall fitness of the trees and negative feedbacks on the growth dynamics of the next season, such
as less buds (Fu et al., 2014;Vander Mijnsbrugge et al., 2016;Crabbe et al., 2016). Although Fu et al. (2014)
suggested a correlation between the bud burst and the onset of autumn leaf senescence, we have found
no relationships for 2018 and 2019 in birch and beech, but a positive relationship in oak (every delay of
one day in the bud burst corresponded to a delay of ± two days in the onset of autumn leaf senescence).
Surprisingly, the onset of autumn leaf senescence did not differ significantly among the different tree
species, which supports the idea that the onset of autumn leaf senescence in different deciduous trees
might be controlled by the same (light related) signal. Perhaps the onset of leaf senescence is timed in a
manner similar to flowering, as put forward by the external coincidence model (i.e. clock-regulated gene
expression and light both determine the perception of photoperiodism) (Böhlenius et al., 2006;Kobayashi
and Weigel, 2007;Koornneef et al., 1991;Yanovsky and Kay, 2002). Other explanations for the lack of
significant differences in the onset of autumn leaf senescence among the species could have been the
small sample size (i.e. eight beech, four birch and four oak trees for the CCI measurements) or the
inaccuracies related to the method of piece-wise linear regressions. Given our results, the drought in 2017,
2018 and 2019 had little impact on the CCI trend and onset of autumn leaf senescence in mature beech,
birch and oak trees.
In this regard, the exact impact of the light quantity and spectral quality on the trigger for the onset of
senescence (directly or indirectly through photoperiodic detection), is not well known in deciduous trees
(Michelson et al., 2018). If phytochrome only responds to the presence of red wavelengths, the effect of
the polycarbonate in the glasshouses must have been minimal. However, experimental biases might be
caused if cryptochrome, which is sensitive to UV light and active at low fluency rates,  played a significant
role in the onset of senescence (Schulze et al., 2019;Smith, 1982). Because very low light intensities are
required by plants to generate a photosynthetic potential (a minimum scalar irradiance of ± 1 μmol/m²)
and very low fluencies (starting from 0.1 μmol/m²) are required for phytochrome action, we assumed the
decrease in the incoming light intensity would not have had a significant effect (Legris et al., 2019;Poorter
et al., 2019;Franklin and Quail, 2010;Legris et al., 2016;Neff et al., 2000;Mancinelli and Rabino, 1978).
Although the onset of autumn leaf senescence in both the tree saplings and the mature trees was not
advanced by drought, heat stress and increased atmospheric aridity, the onset of autumn leaf senescence
in beech saplings was around 22 days earlier than mature beech trees. Such difference could be due to
the different growing conditions (pots versus normal soil), environmental conditions at the different sites,
the difference in the average leaf age (tree saplings have an earlier bud-burst than mature trees) or the
different ecophysiological response of tree saplings and mature trees (e.g. tree saplings are more
vulnerable than mature trees and therefore are likely to use different functional strategies) (Niinemets,
2010;Vander Mijnsbrugge et al., 2016;Pšidová et al., 2015). As there is very little difference in the light
conditions among the different sites, the difference in the day length is unlikely to have affected the
difference in the timing of the onset of autumn leaf senescence between the beech saplings and mature
trees. However, it is possible that the beech saplings have a different sensitivity to the light cues, as they
usually grow in the understory and therefore under a different light regime than mature trees (Brelsford
et al., 2019;Michelson et al., 2018;Chiang et al., 2019).
Concerning the onset of the loss of canopy greenness for all species and opposed to 2017 (i.e. a year with
normal environmental conditions in late-summer and autumn) and 2019 (i.e. a year with high
temperatures in summer, relatively normal precipitation in summer and autumn, but suffering from the
accumulated effects of the rainfall deficit), the onset of the loss of canopy greenness in 2018 was around
three-and-a-half weeks earlier. The canopy greenness metric had been declining earlier in 2018 because
the leaves have likely been shed earlier due to an advanced leaf abscission process to protect the tree
from hydraulic failure (Munné-Bosch and Alegre, 2004;Wolfe et al., 2016). There was also a difference in
the onset of the loss of canopy greenness among the species. This might be due to two reasons. First,
birch (the species with the earliest onset of the loss of canopy greenness) has an indeterministic growth
pattern, which also means continuous leaf mortality. Second, the fact that oak (the species with the latest
onset of the loss of canopy greenness) has typically a second leaf flush, which might connect the difference
between beech and oak to differences in leaf longevity.
Overall, the GAMMs reproduced reliable fits of the CCI and canopy greenness. One of the few observed
issues was a small mismatch between the mean CCI shown by the smoother of the fitted GAMM and the
mean CCI shown by the line plot for the + 3°C treatment at the end of the growing season (early October
– mid November). The overestimation of the CCI in this case might reflect the limitations of using Gaussian
GAMMs here

## 5. Conclusion

The different environmental conditions of three years (comprising a severe dry year and a severe warm
year) did not affect the timing of the onset of autumn leaf senescence in mature beech, birch and oak
forest trees in Belgium. This suggests that deciduous trees have a conservative strategy concerning the
timing of their senescence. Like our mature beech trees, beech saplings exposed to drought, heat stress
and increased atmospheric aridity also did not show any advancement in their onset of autumn leaf
senescence compared to beech saplings in normal conditions. Although the drought, heat stress and
increased atmospheric aridity did not affect the timing of the onset of autumn leaf senescence, it is clear
from our results that they  affect the mortality rate in tree saplings and the leaf mortality in mature trees.

# Appendix A

## Figures

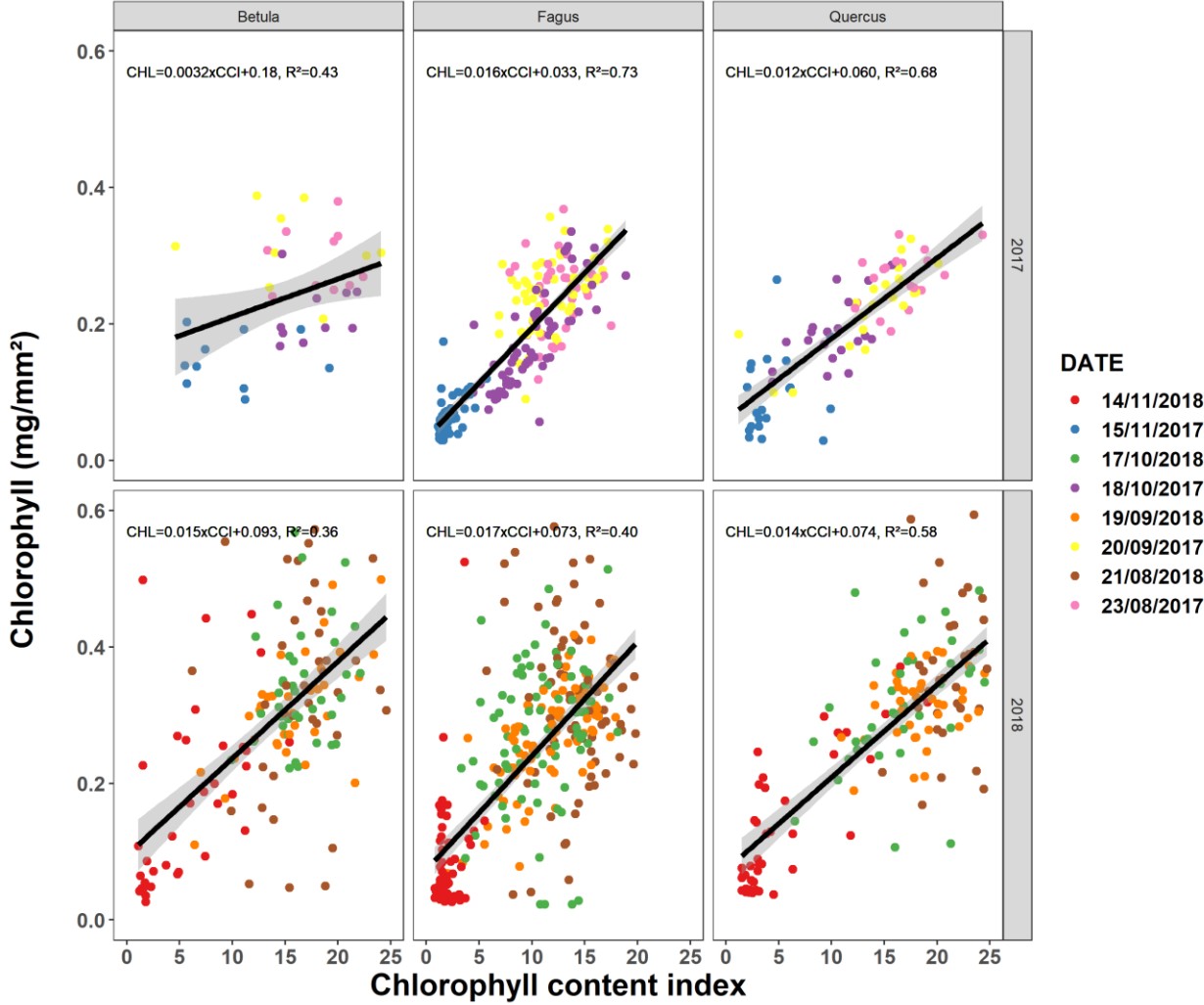


Fig. A1: Relationship between the chlorophyll content index measured using a chlorophyll content meter (CCM-200 plus, Opti-Sciences Inc., Hudson, NH, USA) and the chlorophyll concentration measured using spectrophotometric analysis (Mariën et al., 2019). Between late August and late November 2017-2018, we sampled every month 20-40 leaves (five leaves for four to eight trees) for beech and 10 to 20 leaves (five for two to four trees) for birch and oak. The different colors represent different sampling dates.

720

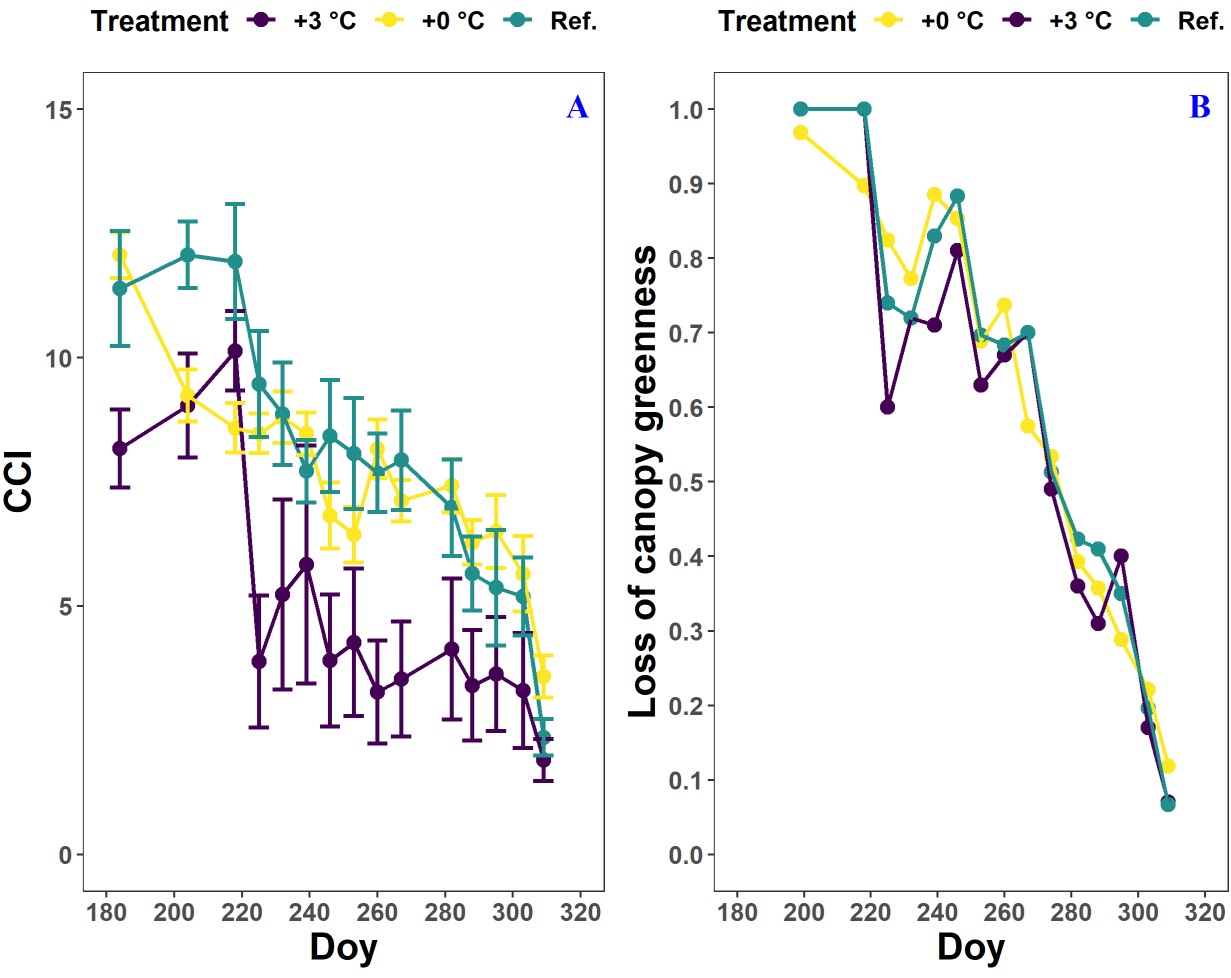

721

Fig. A2: The chlorophyll content index (CCI; panel A) and loss of canopy greenness (panel B) of the *Fagus sylvatica* saplings at the Drie Eiken Campus in Wilrijk for which no breakpoint could be calculated. The dots and error bars represent the mean CCI (panel A) and mean loss of canopy greenness (panel B) with standard errors. The colors represent the CCI or the loss of canopy greenness of the beech saplings in the reference plots (green; Ref.), the glasshouses that followed the outside ambient air temperature (yellow; +0 °C) and the glasshouses that were three degrees warmer than the outside ambient air temperature (purple; +3 °C), respectively.


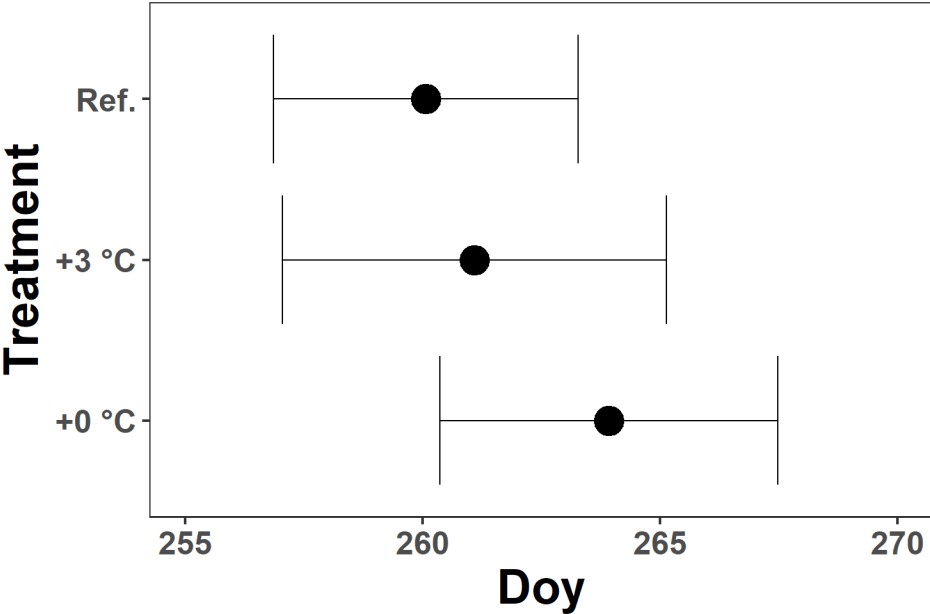


Fig. A3: The mean onset of autumn leaf senescence per treatment for all *Fagus sylvatica* saplings at the
Drie Eiken Campus in Wilrijk. Black dots represent the mean onset of autumn leaf senescence, while the
error bars represent standard errors that indicate the inter-individual variability. All breakpoints are
calculated using the chlorophyll content index data and piecewise-linear regressions ($n_{Ref.}$ = 29; $n_{+0 °C}$ = 26;
$n_{+3 °C}$ = 22). : Ref. represents the breakpoints of the trees in the reference plots, while +0 °C and +3 °C
represents the breakpoints of the trees in the glasshouses under the 'drought/less irrigation' and
'drought/less irrigation, warming and increased atmospheric aridity' treatments, respectively.

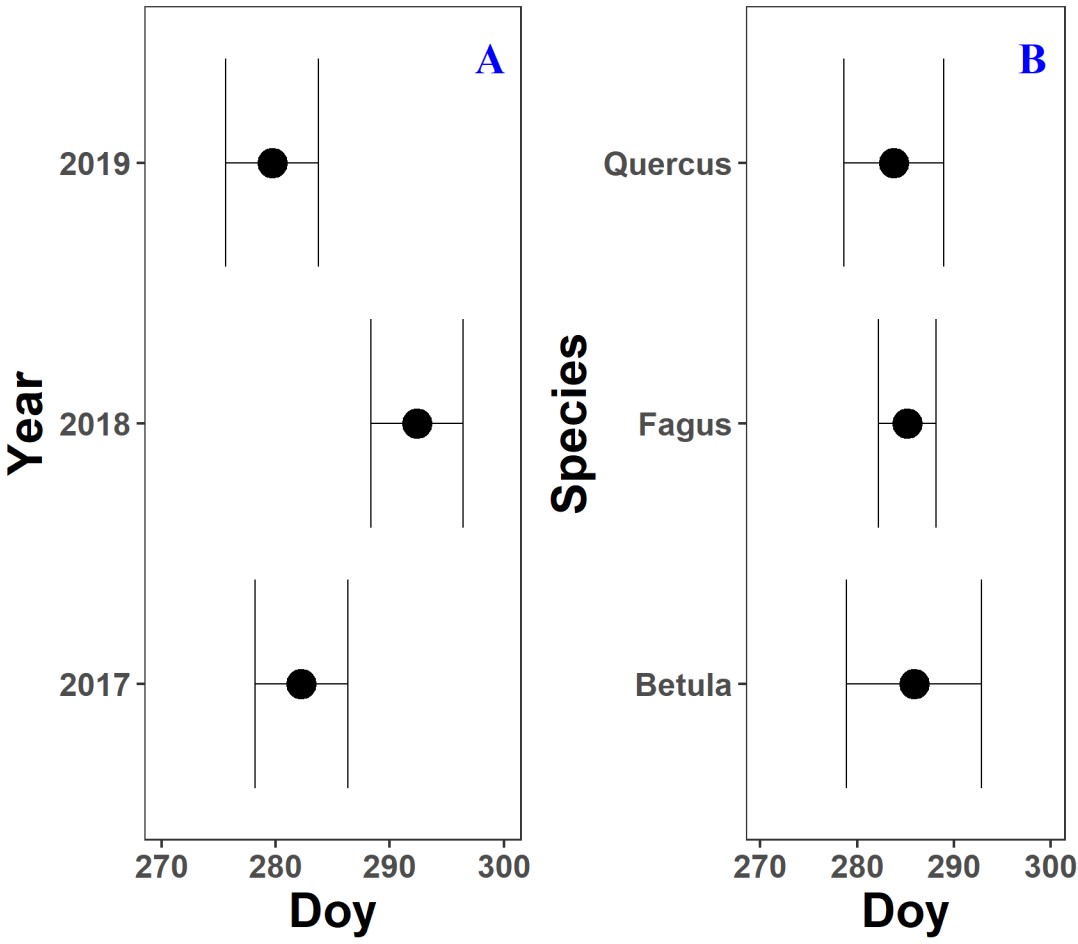


Fig. A4: The mean onset of autumn leaf senescence for three years (panel A; 2017 - 2019) and the three species (panel B; *Fagus sylvatica*, *Betula pendula* and *Quercus robur*) measured on all mature trees at the Klein Schietveld and Park of Brasschaat. Black dots represent the mean onset of autumn leaf senescence, while the error bars represent standard errors that indicate the inter-individual variability. All breakpoints are calculated using the chlorophyll content index data and piecewise-linear regressions ($n_{Fagus}$ = 8; $n_{Betula}$ = 4; $n_{Quercus}$ = 4).

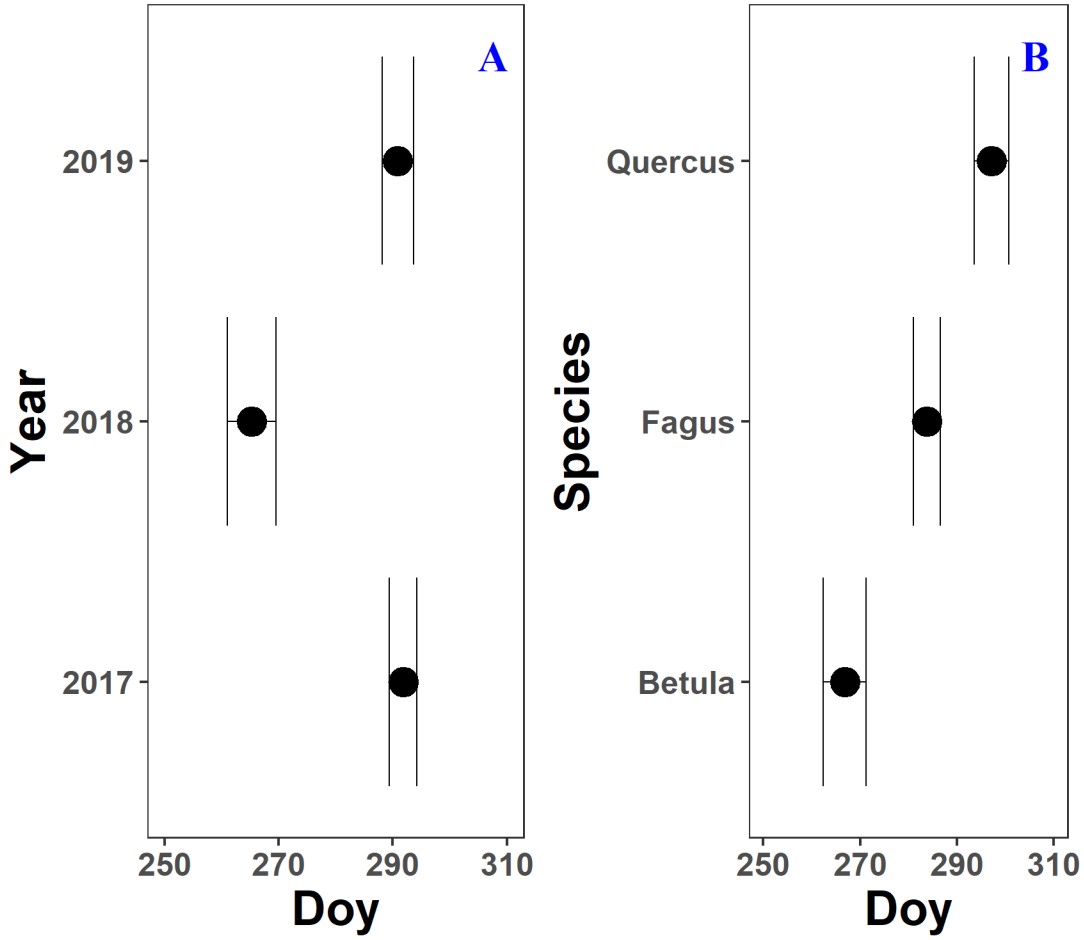

Fig. A5: The mean onset of the loss of canopy greenness for three years (panel A; 2017 – 2019) and the
three species (panel B; *Fagus sylvatica*, *Betula pendula* and *Quercus robur*) measured on all mature trees
at the Klein Schietveld and Park of Brasschaat. Black dots represent the mean onset of the loss of canopy
greenness, while the error bars represent standard errors that indicate the inter-individual variability. All
breakpoints are calculated using the loss of canopy greenness data and piecewise-linear regressions
(nFagus = 16; nBetula = 8; nQuercus = 8).

## Data availability

The code and data corresponding to the work presented in this article is available at Zenodo as doi: 10.5281/zenodo.4559535

## Author contributions

MC and HDB designed the experiment. ID, SL, PW and BM collected the data. PW computed the rainfall deficit, while BM performed all other analyses. BM, PW and MC wrote the text. All authors contributed to the discussions.

## Competing interests

The authors declare that they have no conflict of interest.

## Acknowledgements

The authors acknowledge the funding provided by the ERC Starting Grant LEAF-FALL (714916) and the DOCPRO4 fellowship provided to BM by the University of Antwerp. We also express our gratitude to the Flemish Institute for Nature and Forest (INBO), the Integrated Carbon Observation System (ICOS), the Belgian Royal Meteorological Institute (KMI) and the Royal Dutch Meteorological Institute (KNMI) for providing meteorological data. We would also thank the Agency for Forest and Nature of the Flemish Government (ANB), the Belgian Armed Forces and the Municipality of Brasschaat because they gave permission to conduct research in the study areas. Special thanks are due to Dirk Leyssens (ANB) and Bergen boomverzorging.

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
