# Peer review of "Does drought advance the onset of autumn leaf senescence in temperate deciduous forest trees?"

_Biogeosciences, 2020_

## Short Comment (SC1) · 9 Nov 2020

Various soil parameters are related to the level of water stress in many tree species. The variability in space and time introduced by the interactions due to climate change effects is high and difficult to predict for (sustainable) forest management purposes. However, changes in crown transparency and/or crown coloration may help in order to identify suitable enviromental indicators. What are your findings in this respect?

Please also note the supplement to this comment:
https://bg.copernicus.org/preprints/bg-2020-337/bg-2020-337-SC1-supplement.pdf

**Supplement:**

[supplement omitted: unrelated document]

---

## Author Comment (AC1) · 10 Nov 2020

Dear Mr. Aprile, thank you for your question.

The reader asks whether crown transparency or coloration might be used as an indicator of drought stress for management purposes. It is known that coloration is an indication of stress and that, for instance within a homogenous stand, trees with earlier coloration might be weaker than others (or will be in the future). Even dendrometric parameters (e.g. diameter) do not show this yet. In this way, the practitioner might find it useful to pay attention to trees with earlier coloration. However, the chlorophyll and pigments that make the coloration do not degrade at the same rate during autumn (e.g. see Keskitalo et al. 2005). The rate at which pigments are degraded or increased

during the senescence process is also dependent on many environmental variables (e.g. temperature) and species-specific. Our study focuses on the detection of autumn senescence and how this can be confused by the impact of drought in some deciduous tree species. The loss of canopy greenness (or similar indices) is used to time the onset of senescence. However, at the leaf level, using chlorophyll measurements is a better indicator of the actual nutrient remobilization. The effect of the photoperiod is paramount and specific for the timing of senescence in almost all deciduous tree species. Therefore, particular crown transparency or coloration values will only partly indicate drought stress.

---

## Referee Comment (RC1) · Anonymous Referee #1 · 8 Dec 2020

Summary: Marien and co-authors presented an experimental study in which they assessed the effect of drought stress on leaf senescence for 3 tree species in Belgium forests (mature trees) over 2017-2019 and from manipulative experiments with saplings. Results did not show any effect of drought stress on the timing of leaf senescence. However, the authors observed an effect of drought on the chlorophyll content and the canopy greenness of trees.

Overall, this study is well written. The experimental design is sound, and limits in the protocol and analysis are clearly highlighted and discussed. Results support the conclusions of the manuscript. I don't have major comments on the manuscript, only a few suggestions that might strengthen the analysis:

1) The authors used piecewise logistic regression to estimate the timing of leaf senescence. Some studies suggested that a simple threshold approach leads to better results and maybe a more robust comparison of phenological events. Did the authors tried to compare the timing of senescence using a threshold approach? The absence of observed effect might come from the definition of leaf senescence.

2) Drought stress is defined here as the rainfall deficit. Instead of a meteorological drought index, did the authors tried other physiological drought indices? For example, the ratio of actual over potential evapotranspiration (Stocker et al. 2018) that might be more representative of the stress than the rainfall deficit.

3) Some studies suggested that the timing in leaf unfolding impacts senescence (Fu et al. 2014). Was this effect observed on site? Did the authors include other effects than precipitation, temperature and drought stress in their model? It might be interesting to discuss this point in the discussion section.

4) I suggest the authors to highlight the effect of drought stress on CCI and greenness in the abstract. As they discussed (L. 464), the effect on greenness is probably an important source of confusion in the literature and I think it is an important message of this paper.

I hope the authors will find these comments useful for improving their manuscript.

Best regards,

References:

Stocker, B. D., Zscheischler, J., Keenan, T. F., Prentice, I. C., Peñuelas, J., & Seneviratne, S. I. (2018). Quantifying soil moisture impacts on light use efficiency across biomes. New Phytologist, 218(4), 1430-1449.

Fu, Y. S., Campioli, M., Vitasse, Y., De Boeck, H. J., Van den Berge, J., AbdElgawad, H., ... & Janssens, I. A. (2014). Variation in leaf flushing date influences autumnal senescence and next year's flushing date in two temperate tree species. Proceedings of the National Academy of Sciences, 111(20), 7355-7360.

---

## Short Comment (SC2) · 8 Dec 2020

Certainly, water stress and/or related soil parameters cannot explain all the time variability in foliage coloration. However, it can contribute and therefore depict reaction of vegetation at both the small parcel and landscape level. I tested it in two works (please see the attached publications). This does not mean to contrast the authors' work but providing them with some more information that might be useful.

And, I would also like to attract the attention on the fact that crown transparency and coloration/withering, which have frequently been used as parameters related to water stress due to both direct and indirect causes, may not necessarily be indicative of the level of the humidity content of the tree and/or health status.

[Figure]

Please also note the supplement to this comment:
https://bg.copernicus.org/preprints/bg-2020-337/bg-2020-337-SC2-supplement.zip
* * *

---

## Referee Comment (RC2) · Anonymous Referee #2 · 17 Dec 2020

The article analyzes the impact of drought on the onset of autumn senescence and the difference featured by different temperate deciduous tree species. The authors used a manipulative experiment of tree beech sapling and three years of data on beech, birch, and oak trees. The authors show that drought did not affect the onset of senescence. Tree saplings showed high mortality with drought, and mature trees showed higher leaf mortality. No significant differences across species were observed. The manuscript deals with a significant subject, senescence, about which not much is yet known. Understanding the senescence process, particularly in relation to drought, is fundamental to predict the phenological cycle of temperate trees better.

Regarding the greenhouse experiment, I have a methodological concern. From the data reported in Fig 1 seems that the "drought" treatment does not have a significant

(should be tested statistically thought) effect on soil moisture (Fig 3c). Instead, the effect was mainly an increase of VPD that is not drought but an increase of the atmospheric evaporative demand. One of the factors linked to the earlier senescence in the case of drought is abscisic acid accumulation (ABA). Long term ABA responses should be more induced by soil moisture. Root perceives reducing soil moisture and upregulate ABA synthesis. ABA is a factor controlling earlier senescence (and stomatal regulation). I am not aware of studies showing the high VPD can trigger the same response in terms of upregulation of ABA. It could be that the lack of response observed was simply de to the fact that the reduction of soil moisture was not enough to trigger the physiological response inducing earlier senescence. Also, in general, I would not call drought the treatment. Given the data shown in Figure 1, I think it is more heat stress. Please provide more insights to understand whether the treatment can be indeed called drought treatment. If not, I would suggest talking about heat stress and increased atmospheric aridity. This would not diminish the paper. There is a lot of discussion on the different repose of plants to decreasing soil moisture and/or increasing VPD, and here I think the authors are looking at increased VPD and not necessarily at drought. This can also support the discussion of the differences between 2018 (more soil moisture stress) and 2019 (more heat and VDP stress) – see discussion at line 469-470. The 20% reduction of incoming light should also be better addressed (Line 162-163): though unclear, it seems that senescence is controlled by photoperiod. How does 20% - decrease in incoming radiation affect the photoperiod? The authors should check this and evaluate if the reduction of light has an impact on the results.

In the methods section 2.1.2, when the CCI is mentioned the first time, I expected a description of the sampling (that comes later). I think it would be beneficial to move section 2.2 above, where the CCI is mentioned the first time. Preferentially, put a reference in paragraph 2.1.2 to paragraph 2.2. The meteo stations are 20 and 60 km from the sites. But there is no information about where these stations were located (in a city, in a forest, in a grassland, at which height). Even if the climate regimes can be similar at a distance of 20-60 km can we be sure that the microclimate is comparable?

At the moment all this info are missing. I suggest the authors carefully check all this information and provide a methods description that can prove the study's robustness.

1) The equation and symbols do not follow the scientific format. I suggest to rewrite them. Also many variables have names that are more for a programming language but not following the scientific notion. I suggest to follow the IUPAC standards, or at least try to go close to that format. Avoid using "Leaf_place"Also please define the variable the first time is used, and then stick with the symbol: one example is the "day of the year" that in the equation 2 (model 1) is Doy and in the text is "day of the year"

2) If I am not wrong there is a mistake in Eq 1. First if rH should not be expressed in % as indicated but as fraction (rH[%]/100)

Here the result of VPD with the current equation > T <- 25 > rH <- 50 > e0 <- 613.75*exp((17.502*T)/(240.97+T)) > e <- rH*e0 > VPD <- e0-e > VPD [1] -155829.6

Moreover, even if the rH is used in the correct unit, the VPD unit is wrong. The resulting VPD from this equation is in Pa and not kPa as indicated at line 144.

> T <- 25 > rH <- 50/100 > e0 <- 613.75*exp((17.502*T)/(240.97+T)) > e <- rH*e0 > VPD <- e0-e > VPD [1] 1590.098

The VPD reported in the figures seems correct, therefore please verify is there is a problem in the Equation.

3) There are few track changes and typos in the manuscript. Please edit careful the article a. Line 249, line 415, 416, 417, 4) The reference to the R package is a bit strange R/ggpubr etc. Please modify in: "we use the R package ggpubr (Reference)". But it is very nice that the authors cite all the packages. This is important and often overlooked. 5) Please report "Model 1 and 2" in a less R script style. Please use mathematical notation 6) 6) I think the breakpoint analysis was achieved with the "segmented" package and not dplyr", correct? 7) Line 464-465 – this is interesting, please elaborate more this point.

---

## Author Comment (AC3) · 12 Jan 2021

Dear Anonymous Referee 1,

Thank you for your review and suggestions. We will respond here to your comments:

1) The Referee asks whether we considered alternatives (e.g. simple thresholds in canopy coloration percentage) to the piecewise linear regressions to determine the timing of the onset of leaf senescence. We are aware that different methodologies (e.g. from simple thresholds to complex network-based approaches) can, and are, used to estimate the timing of leaf senescence. In fact, we compared the results obtained using piecewise linear regressions and 50% canopy coloration/ leaf fall thresholds (i.e. assuming that the onset of leaf senescence can be approximated with the timing when

50% of the canopy lost the green color) in previous work, and showed that the methods provide different results, with 50% thresholds giving results that are consistently later (Mariën et al., 2019). We agree that comparing different methods might nevertheless yield advantages, as the timing of leaf senescence is inherently a problem of deriving a trend in complex ecological data (i.e. data that is, for example, hierarchical and non-linear). Exactly for deriving this trend, and as an extra regression method to compare to the piecewise linear regressions, we used the generalized additive mixed models and plotted the resulting factor-smooth interaction smoothers with 95% simultaneous confidence intervals.

2) The Referee asks whether we considered different physiological drought indices (e.g. the ratio of actual over potential evapotranspiration as in Stocker et al. (2018)). We agree that other indices would be useful. However, calculating the index proposed in Stocker et al. (2018) would not be feasible in a short term. An additional difficulty is that these calculations would require a hydrological model and are strongly dependent on local soil characteristics. Furthermore, most local meteorological stations do not provide evaporation data. Finally, note that long-term values of the rainfall deficit, as reported in Fig. 3, are rather exceptional. Therefore, the drought stress index that is reported here should be sufficiently representative for our purposes. Note that we actually do not use the drought index in our calculations or models but only use it to describe the meteorological conditions within the three year study period.

3) The Referee asks whether we observed an effect of the timing of the leaf unfolding on the senescence timing and whether we considered including other variables into our model. The age of leaves might indeed affect the timing of the onset of senescence, especially in species with an indeterministic growth pattern (e.g. birch). Therefore, we will test the correlation between leaf unfolding and senescence timing (some preliminary results are available in the supplementary file 'TEST_BB_OLS_markdown'). However, our dataset will be limited to mature trees in 2018 and 2019, as spring data for 2017 are not available and leaf unfolding for the trees in the manipulative experiment was

affected by establishment effects. However, note that we did not follow the exact same leaves from bud burst to senescence. In addition, it is hard to disentangle whether the different timing of the bud burst affects the timing of leaf senescence, or whether the opposite is the case (Marchand et al 2020). Our models simply included "treatment", "leaf position", "day of the year" and "individual_tree" for the manipulative experiment, and "year", "species", "leaf position", "day of the year" and "individual_tree" for mature trees. So, they did not include meteorological variables. A significant upgrade in the modeling work is the application of GAMM or GAMLSS models, where correlations between e.g. seasonal chlorophyll data and meteorological data, are accounted for. The amount of work in applying these models to senescence trends is significant and we are working on this in the next manuscript.

4) The referee suggests highlighting the effect of drought stress on CCI and greenness in the abstract, as discussed on L. 464 ("For the mature trees, the different drought response of the autumn pattern of chlorophyll (no effect) and the loss of canopy greenness (advanced and enhanced) is probably an important reason of confusion still present today in the literature on the relationship between drought and autumn senescence"). We thank the referee for this suggestion and will consider this in the revision.

Kind regards,

The authors

Please also note the supplement to this comment:
https://bg.copernicus.org/preprints/bg-2020-337/bg-2020-337-AC3-supplement.pdf

**Supplement:**

**TEST_BB_OLS**

Bertold Mariën

7-1-2021

```r
###############################################################
###############################################################
data <-
read.csv("C:/Users/BMarien/Desktop/laptop/Own_work/Droughts_do_not_advance_th
e_onset_of_autumn_leaf_senescence_in_temperate_deciduous_forest_trees/Data/Da
ta_used_in_article/BB_vs_CCI_breakpoints.csv", sep=";")
names(data)

**[1] "ID"      "Site"    "Species" "Number"  "Year"    "BB"      "SE_BB"**
**[8] "OLS"     "SE_OLS"**

ID2 <- as.factor(as.character(data[,1]))
SITE2 <- as.factor(as.character(data[,2]))
SPECIES2 <- as.factor(as.character(data[,3]))
NUMBER2 <- as.factor(as.character(data[,4]))
YEAR2 <- as.numeric(as.character(data[,5]))
BB2 <- as.numeric(as.character(data[,6]))
OLS2 <- as.numeric(as.character(data[,7]))

##
data3 <- data.frame(ID2, SITE2, SPECIES2, NUMBER2, YEAR2, BB2, OLS2)

##
data4 <- na.omit(data3)

library(mgcv)

**Loading required package: nlme**

**This is mgcv 1.8-31. For overview type 'help("mgcv-package")'.**

m1 <- gamm(OLS2 ~ s(BB2, bs = 'tp', by = SPECIES2),
         random =list(NUMBER2=~1),
         family = gaussian(),
         method = "REML",
         data = data4)

**Get the summary results**
summary(m1$gam)

##
**Family: gaussian**
**Link function: identity**
```

```
* * *
**Formula:**
**OLS2 ~ s(BB2, bs = "tp", by = SPECIES2)**
* * *
**Parametric coefficients:**
**Estimate Std. Error t value Pr(>|t|)**
**(Intercept)    8.486      1.236   6.863 5.64e-07**
**---**
**Signif. codes:  0 '***' 0.001 '**' 0.01 '*' 0.05 '.' 0.1 ' ' 1**
* * *
**Approximate significance of smooth terms:**
**edf Ref.df      F  p-value**
**s(BB2):SPECIES2Betula  1.000  1.000  2.592    0.121**
**s(BB2):SPECIES2Fagus   1.690  1.690  4.747    0.079 .**
**s(BB2):SPECIES2Quercus 5.531  5.531 23.976 1.35e-12**
**---**
**Signif. codes:  0 '***' 0.001 '**' 0.01 '*' 0.05 '.' 0.1 ' ' 1**
* * *
**R-sq.(adj) =  0.842**
**Scale est. = 15.721    n = 32**

**Get a quick & dirty plot**
plot(m1$gam, rug = TRUE, all.terms = TRUE, #pages = 1,
     seWithMean = TRUE, shade = TRUE, shade.col = "hotpink", # SE of partial
effect + SE of model intercept, reflect overal uncertainty better than CI
     shift = coef(m1$gam)[1]) #Shift scale to include intercept
```

[Figure]

[Figure]

[Figure]

```
**Check the assumptions of normality & homogeneity quick & dirty**
par(mfrow = c(2,2))
gam.check(m1$gam)
```

[Figure]

[Figure]

[Figure]

```
* * *
**'gamm' based fit - care required with interpretation.**
**Checks based on working residuals may be misleading.**
**Basis dimension (k) checking results. Low p-value (k-index<1) may**
**indicate that k is too low, especially if edf is close to k'.**
* * *
**k'   edf k-index p-value**
**s(BB2):SPECIES2Betula  9.00 1.00    1.08    0.58**
**s(BB2):SPECIES2Fagus   9.00 1.69    1.08    0.61**
**s(BB2):SPECIES2Quercus 9.00 5.53    1.08    0.61**
```

---

## Author Comment (AC4) · 12 Jan 2021

Dear Anonymous Referee 2,

Thank you for your review and suggestions. We will respond here to your comments:

1) The Referee asks whether it is possible that the reduction of soil moisture in the glasshouse experiment was not enough to trigger the physiological response inducing earlier senescence. He therefore questions whether an increased VPD can trigger the upregulation of ABA. Literature shows that ABA, which is known to control earlier senescence, is indeed upregulated as a response to the stomatal changes corresponding to changing vapor pressure deficit levels (McAdam and Brodribb, 2016;McAdam et al., 2016;Bauerle et al., 2004;Xie et al., 2006). We agree that the treatments

[Figure]

+0ËŽC and +3ËŽC did not result in large differences in soil water content. However, we will test this statistically, as suggested (for example, see the supplementary file 'TEST_SWC_markdown' and Rose et al. (2012) for additional information on the possible statistical methodology). On the other hand, it is likely that larger differences were present between the reference plots and the treatments, as the reference plots were irrigated more (L. 160), and such irrigation regime showed values of soil water content of up to ca. 0.20 $m^3$/m while the values of 0.05 $m^3$/$m^3$ were reached in the treatments (see Fig 1; unfortunately, sensor malfunctioning did not allow us to gather soil water content data for the reference plots). Given that we observed a high mortality, it might have been the case that our +3 ËŽC treatment was too extreme, triggering necrosis instead of earlier senescence (Munné-Bosch and Alegre, 2004).

2) The Referee suggests we talk about heat stress rather than drought stress. As mentioned above, the reference plots were irrigated more than the treatments plots (L. 148 - 149; 159 - 160). Therefore, the more appropriate definition would be "treatment based on warming, less irrigation and increased atmospheric aridity". We could use this definition (although longer and somewhat impractical it is the closest to reality). In reference to L. 469 - 470 ("...the drought of 2019, which coincided with several heat waves, might have been less damaging for late summer leaf dynamics, than the drought of 2018..."), a more detailed comparison between experimental manipulation and mature trees in years 2018 and 2019 would have required a factorial approach separating drought and warming, while our design was more basic. In addition, as shown in figure 3, the rainfall deficit was high in all years. It is true that the rainfall deficit was extremely high in 2018 – 2019, but the rainfall deficit was also high in 2017 – 2018 and 2019 – 2020. Likely, more site specific measurements on the soil water content would indeed have been useful. Note, however, that figure 2 and table 1 also indicate that there was not only little precipitation but also that this precipitation fell in irregular patterns, making potential droughts more likely.

3) The Referee asks to comment on the effect of the 20% reduction in light due to

the colorless polycarbonate roof in the glasshouses (L. 162 – 163; "A draw-back of the experiment is that the saplings in the reference plots received more incoming light (i.e. $\pm$ 20%) than the saplings in the glasshouses (Van den Berge et al., 2011)"). The Reviewer raises an interesting point: can a reduction / change in the light affect the photoperiod? Preliminary tests suggested that the ratio of light in different wavelengths (e.g. R/FR) during civil twilight (i.e. what is required for phytochrome to detect the photoperiod) does not change seasonally significantly in our study area. This provide indirect evidence for us to believe that our light reduction (limited to 20%), combined with the fact that very low light intensities are needed for plants to detect photoperiod (Legris et al., 2019;Poorter et al., 2019;Franklin and Quail, 2010), would not have caused significant changes in photoperiod. We agree that it could be interesting to test the effect of the roof alone. However, this is not feasible in the short term. The effect of the roof is also partly captured by the results on the saplings in the +0 °C treatment glasshouses.

4) The Referee asks to consider restructuring section 2.2 and 2.1.2. We will consider this in the revision.

5) The Referee asks to provide more information on the meteorological stations. We will add the following information to the manuscript in the revision. (1) The station of Ukkel is located within a green area in the suburb of Brussels (thus, classifiable as "urban park"). The microclimate is expected to be different than at our study sites. However, data from Ukkel were used to describe the intra-annual variability and long-term trends (Table 1 and Fig. 3), which are less affected by microclimate. (2) The meteorological station of Brasschaat is very close to our sampling site in the Park of Brasschaat and in the Klein Schietveld ($\pm$ 3 km and $\pm$ 4 km, respectively). The meteorological station in Brasschaat is a 40 m high scaffolding tower, at which measurements are taken at various heights, and stands in a patch of mixed forest covered mainly by Scots pines and deciduous tree species, such as oak and birch (see Carrara et al. (2003) for more information). Data of the temperature, precipitation and humidity were

taken at the top of the tower. Data from Brasschaat were used to describe the seasonal pattern in 2017, 2018 and 2019, and as input to the models. (3) The station of Woensdrecht is located in an open field at a local airport surrounded by heathland and urban area. It is located near the Markiezaatsmeer, an enclosed swamp ecosystem, within the river mouth of the Schelde. The measurements in both Ukkel and Woensdrecht are taken at a height of 1.5 m. However, these data were only used as gap-filling in case of short term gaps in the long-term Brasschaat series. In terms of differences in the microclimate, it is indeed not ideal that we needed to use data from the meteorological stations of Ukkel and Woensdrecht. However, we are limited here by the availability of the data and the meteorological stations of Ukkel and Woensdrecht are closest (and most representative) for our sampling sites.

6) The Referee comments on the style of the model notation and suggests to better define the variables at first use. We will define the variables further at first use and avoid inconsistencies. However, both the descriptive style and mathematical notation are based on examples and suggested notation in the specific literature (Zuur et al., 2007;Zuur et al., 2010;Zuur et al., 2011;Zuur et al., 2016;Simpson, 2018;Pedersen et al., 2019;Wood, 2017) and readers interested in background references might find it easier if style consistency is respected. Perhaps the Editor can comment on the journals preference?

7) The Referee notes there is an error in the units of the equation on the vapor pressure deficit. Thanks, we will correct this in the revision. The actual and saturation vapor pressure deficit are indeed in Pa, while the relative humidity should be noted as a fraction. The data was indeed calculated using the correct equation.

8) The Referee points out some typo's. The will be addressed in the revision.

9) The Referee suggests writing R packages in a different format. If preferred by the Editor, we will address this in the revision.

10) The Referee suggests using only the mathematical notation for model 1 and 2.Considering the literature (see the response on comment 6) and the preference of the Editor, we will address this in the revision.

11) The Referee suggests to remove the reference to the R package "DPLYR" as the breakpoint analysis is done only using the R package "SEGMENTED". While "DPLYR" was used for data wrangling, we agree "SEGMENTED" is indeed the package that is used for the breakpoint analysis. We will remove the reference to "DPLYR" in the revision.

12) The referee asks to elaborate on L. 464. ("For the mature trees, the different drought response of the autumn pattern of chlorophyll (no effect) and the loss of canopy greenness (advanced and enhanced) is probably an important reason of confusion still present today in the literature on the relationship between drought and autumn senescence"). We thank the referee for this suggestion and will consider this in the revision. While the detoxification of chlorophyll is a prerequisite for the expression of different coloration values, chlorophyll does not degrade at the same speed as other leaf pigments. In fact, not even all leaf pigments degrade (or are formed) at the same velocity throughout the senescence process (Keskitalo et al., 2005). Consequently, observations of changing coloration levels are difficult to interpret. Moreover, note that coloration measurements also take into account leaf yellowing and mortality due to hydraulic failure.

Kind regards, The authors

Please also note the supplement to this comment:
https://bg.copernicus.org/preprints/bg-2020-337/bg-2020-337-AC4-supplement.pdf

**Supplement:**

**TEST_SWC**

Bertold Mariën

7-1-2021

```r
data <-
read.csv2("C:/Users/BMarien/Desktop/laptop/Own_work/Droughts_do_not_advance_t
he_onset_of_autumn_leaf_senescence_in_temperate_deciduous_forest_trees/Data/D
ata_used_in_article/W_exp_NSWC_2018NA.csv")
names(data)
```

```
**[1] "number"         "Date"              "X"              "DOY"**
**[5] "Time"           "PM"                "Chamber"        "Treatment"**
**[9] "water.content"  "water.content_NA"**
```

```r
##
DOY <- as.numeric(as.character(data[,4]))
CHAMBER <- as.factor(as.character(data[,7]))
TREATMENT <- as.factor(as.character(data[,8]))
SWC <- as.numeric(as.character(data[,10]))
CHAMBER <- as.factor(as.character(data[,7]))

##
data1 <- data.frame(DOY, SWC, TREATMENT, CHAMBER)

##
data1 <- na.omit(data1)
names(data1)
```

```
**[1] "DOY"        "SWC"        "TREATMENT" "CHAMBER"**
```

```r
**Test GAM**
library(mgcv)
```

```
**Loading required package: nlme**
```

```
**This is mgcv 1.8-31. For overview type 'help("mgcv-package")'.**
```

```r
m1 <- gamm(SWC ~ s(DOY, bs = 'tp', by = TREATMENT),
           family = gaussian(),
           method = "REML",
           data = data1)

**Get the summary results**
summary(m1$gam)
```

```
##
**Family: gaussian**
```

```
**Link function: identity**
##
**Formula:**
**SWC ~ s(DOY, bs = "tp", by = TREATMENT)**
##
**Parametric coefficients:**
**Estimate Std. Error t value Pr(>|t|)**
**(Intercept) 0.1652966  0.0001259    1313   <2e-16**
**---**
**Signif. codes:  0 '***' 0.001 '**' 0.01 '*' 0.05 '.' 0.1 ' ' 1**
##
**Approximate significance of smooth terms:**
**edf Ref.df    F p-value**
**s(DOY):TREATMENT+0 °C 8.919  8.919 5105  <2e-16**
**s(DOY):TREATMENT+3 °C 8.974  8.974 8373  <2e-16**
**---**
**Signif. codes:  0 '***' 0.001 '**' 0.01 '*' 0.05 '.' 0.1 ' ' 1**
##
**R-sq.(adj) =  0.514**
**Scale est. = 0.0018061  n = 113912**

**Get a quick & dirty plot**
plot(m1$gam, rug = TRUE, all.terms = TRUE, #pages = 1,
     seWithMean = TRUE, shade = TRUE, shade.col = "hotpink", # SE of partial
effect + SE of model intercept, reflect overal uncertainty better than CI
     shift = coef(m1$gam)[1]) #Shift scale to include intercept
```

[Figure]

[Figure]

```
**Check the assumptions of normality & homogeneity quick & dirty**
par(mfrow = c(2,2))
gam.check(m1$gam)
```

**Normal Q-Q Plot**

[Figure]

**Resids vs. linear pred.**

**Histogram of residuals**

[Figure]

**Response vs. Fitted Values**

```
##
**'gamm' based fit - care required with interpretation.**
**Checks based on working residuals may be misleading.**
**Basis dimension (k) checking results. Low p-value (k-index<1) may**
**indicate that k is too low, especially if edf is close to k'.**
##
**k'  edf k-index p-value**
**s(DOY):TREATMENT+0 °C 9.00 8.92     0.8  <2e-16**
**s(DOY):TREATMENT+3 °C 9.00 8.97     0.8  <2e-16**
**---**
**Signif. codes:  0 '***' 0.001 '**' 0.01 '*' 0.05 '.' 0.1 ' ' 1**

**ACF and pACF**
layout(matrix(1:2, ncol = 2))
acf(resid(m1$lme, type = "normalized"),  main = "ACF")
pacf(resid(m1$lme, type = "normalized"),  main= "pACF")
```

[Figure]

```
layout(1)

library(forecast)

**Registered S3 method overwritten by 'quantmod':**
**method            from**
**as.zoo.data.frame zoo**

##
**Attaching package: 'forecast'**

**The following object is masked from 'package:nlme':**
##
**getResponse**

arma_res2 <- auto.arima(resid(m1$lme, type = "normalized"),
                        approximation=TRUE)
arma_res2$coef

**ma1         ma2         ma3         ma4         ma5**
**-0.06184429 -0.03337808 -0.02644633 -0.02334412 -0.02144040**

**plot**
library(gratia)
m2_ci_sim <- confint(m1,
                     level = 0.95,
                     parm = "DOY",
                     type = 'simultaneous',
```

```
                    nsim = 10000,
                    n = 200,
                    shift = TRUE)
names(m2_ci_sim)

**[1] "smooth"       "by_variable" "DOY"          "est"          "se"**
**[6] "TREATMENT"    "crit"         "lower"        "upper"**

head(m2_ci_sim)

**# A tibble: 6 x 9**
**smooth          by_variable  DOY    est       se TREATMENT  crit  lower**
upper
**<chr>           <fct>        <dbl> <dbl>   <dbl> <fct>      <dbl> <dbl>**
<dbl>
**1 s(DOY):TREATME~ TREATMENT     145  0.0496 1.06e-3 +0 °C      2.99 0.0464**
0.0527
**2 s(DOY):TREATME~ TREATMENT     146. 0.0522 9.84e-4 +0 °C      2.99 0.0492**
0.0551
**3 s(DOY):TREATME~ TREATMENT     147. 0.0548 9.09e-4 +0 °C      2.99 0.0521**
0.0575
**4 s(DOY):TREATME~ TREATMENT     148. 0.0574 8.37e-4 +0 °C      2.99 0.0549**
0.0599
**5 s(DOY):TREATME~ TREATMENT     149. 0.0600 7.69e-4 +0 °C      2.99 0.0577**
0.0623
**6 s(DOY):TREATME~ TREATMENT     150. 0.0626 7.07e-4 +0 °C      2.99 0.0605**
0.0647

**plot1**
library(ggplot2)
plot1 <- ggplot(data = m2_ci_sim, aes(x= DOY, y = est,
                                      color = smooth,
                                      fill=smooth)) +
  geom_ribbon(data = m2_ci_sim,
              aes(ymin = lower, ymax = upper, x = DOY), # simultaneous
              alpha = 0.3) +
  geom_line(size = 1.2) +
  scale_x_continuous(name = "DOY",
                     breaks =  seq(145,320,20),
                     limits = c(145,320)) +
  scale_y_continuous(name = "SWC (m³/m³)",
                     breaks =  seq(0,0.25,0.05),
                     limits = c(0,0.25)) +
  annotate("rect", xmin=121,xmax=181, ymin=-Inf,ymax=Inf, fill='lightblue',
alpha = 0.4)

##
library(viridis)

**Loading required package: viridisLite**
```

```
vir_col <- viridis(2, option = 'D')
cols <- c("s(DOY):TREATMENT+0 °C" = vir_col[1],
          "s(DOY):TREATMENT+3 °C" = vir_col[2])
labels <- c("s(DOY):TREATMENT+0 °C" = "+0°C",
            "s(DOY):TREATMENT+3 °C" = "+3°C")

plot1 <- plot1 +
  scale_color_manual(name = "Treatment",
                     values = cols,
                     labels = labels) +
  scale_fill_manual(name = "Treatment",
                    values = cols,
                    labels = labels)

plot1 <- plot1 +
  theme_bw() +
  theme(panel.grid.major = element_blank(),
        panel.grid.minor = element_blank(),
        axis.title.x = element_text(size=18, face="bold"),
        axis.text.x = element_text(size=12, face="bold"),
        axis.title.y = element_text( size=18, face="bold"),
        axis.text.y = element_text(size=12, face="bold"),
        legend.title = element_text(size=14, face="bold"),
        legend.text = element_text(size=12, face="bold"),
        strip.text = element_text(size=12, face = "bold"),
        legend.position ="right")
plot1

**Warning: Removed 52 row(s) containing missing values (geom_path).**

**Warning: Removed 1 rows containing missing values (geom_rect).**
```

[Figure]

```
################################################################
################################################################

**Compare the smooths following Rose et al. (2012)**
pdat <- expand.grid(DOY = seq(145, 320, length = 175),
                    TREATMENT = c("+0 °C","+3 °C"))
xp <- predict(m1$gam, newdata = pdat, type = 'lpmatrix')

##
c1 <- grepl('+0 °C', colnames(xp))
c2 <- grepl('+3 °C', colnames(xp))

##
r1 <- with(pdat, TREATMENT == '+0 °C')
r2 <- with(pdat, TREATMENT == '+3 °C')

##
X <- xp[r1, ] - xp[r2, ]

##
X[, ! (c1 | c2)] <- 0

##
X[, !grepl('^s\\(', colnames(xp))] <- 0

##
```

```r
dif <- X %*% coef(m1$gam)

##
se <- sqrt(rowSums((X %*% vcov(m1$gam, unconditional = TRUE)) * X))

##
crit <- qt(.975, df.residual(m1$gam))
upr <- dif + (crit * se)
lwr <- dif - (crit * se)

##
smooth_diff <- function(model, newdata, f1, f2, var, alpha = 0.05,
                        unconditional = FALSE) {
  xp <- predict(model, newdata = newdata, type = 'lpmatrix')
  c1 <- grepl(f1, colnames(xp))
  c2 <- grepl(f2, colnames(xp))
  r1 <- newdata[[var]] == f1
  r2 <- newdata[[var]] == f2

  ##
  X <- xp[r1, ] - xp[r2, ]

  ##
  X[, !(c1 | c2)] <- 0

  ##
  X[, !grepl('^s\\(', colnames(xp))] <- 0
  dif <- X %*% coef(model)
  se <- sqrt(rowSums((X %*% vcov(model, unconditional = unconditional)) * X))
  crit <- qt(alpha/2, df.residual(model), lower.tail = FALSE)
  upr <- dif + (crit * se)
  lwr <- dif - (crit * se)
  data.frame(pair = paste(f1, f2, sep = '-'),
             diff = dif,
             se = se,
             upper = upr,
             lower = lwr)
}

##
comp1 <- smooth_diff(m1$gam, pdat, '+0 °C', '+3 °C', 'TREATMENT')
comp <- cbind(Doy = seq(145, 320, length = 175),
              rbind(comp1)) #comp2, comp3

##
library(ggplot2)
plot2 <- ggplot(comp, aes(x = Doy, y = diff)) +
  geom_ribbon(aes(ymin = lower, ymax = upper), alpha = 0.2) +
  geom_line(size = 1) +
```

```
  coord_cartesian(ylim = c(-0.05,0.05)) +
  geom_hline(yintercept=0,linetype="dashed", color = "red", size = 1) +
  labs(x = 'DOY', y = 'Difference in \n SWC trend (m³/m³)') +
  scale_x_continuous(name = "DOY",
                     breaks =  seq(145,320,20),
                     limits = c(145,320)) +
  annotate("rect", xmin=145,xmax=181, ymin=-Inf,ymax=Inf, fill='lightblue',
alpha = 0.4)

plot2_F <- plot2 +
  theme_bw() +
  theme(panel.grid.major = element_blank(),
        panel.grid.minor = element_blank(),
        axis.title.x = element_text(size=18, face="bold"),
        axis.text.x = element_text(size=12, face="bold"),
        axis.title.y = element_text( size=18, face="bold"),
        axis.text.y = element_text(size=12, face="bold"),
        legend.title = element_text( size=14, face="bold"),
        legend.text = element_text(size=12, face="bold"),
        legend.position ="bottom")
plot2_F
```

---

## Author Response (AR1)

**Note:** Comments by the Editor and Reviewers are presented in red, while the first (i.e. initial response)
and second responses (i.e. lines of changes that have been made in the manuscript) of the authors are
given in black and blue, respectively.

 **By the Editor**

Thanks for submitting your work to Biogeosciences. This manuscript was read and commented on by two reviewers as well as a non-assigned reader; in general all provide thoughtful, interesting comments and useful suggestions. I have read and reviewed this feedback, the authors' responses, and of course the manuscript itself.

In general I agree with the reviewers that this is fundamentally a strong, interesting manuscript, and your responses are thoughtful and adequate. I do also, however, share some of their concerns about the experiments treatments and how they're described. In particular, R2's concern about heat stress being confounded with drought should be carefully considered, and the potential problem (or not) of different light levels comprehensively discussed. The comments about notation and R package descriptions are well taken although optional from my point of view; the critical thing is that things are clearly documented. On that note, the availability of your data and analytical code needs to be clearly specified, ideally with a link to a permanent repository.

In summary, this is an interesting manuscript that needs moderate to major revisions before further consideration.

*Dear Editor,*

*Thank you for your suggestions and comments.*

*We have improved the description of the treatments used in our sapling experiments (L. 161 - 179), adapted our terminology of the treatment effects (L. 171 – 179; L. 178; L. 491; L. 580; L. 585; L. 588: L. 620; L. 662 and L 691 – 693) and improved the discussion about the treatments (L. 589 - 594). We also added data on the volumetric soil water content measured at the flux tower in Brasschaat (see L. 204; fig. 2; L. 219 – 231 and L. 244 -252) and commented on this data (L. 477 – 479 and L. 612 - 618). Significant effort has been made to clarify and discuss the (potential) effect of the polycarbonate roof in L. 125 – 126; L. 184 – 189 and L. 650 – 659), although knowing the exact effect would require a different experimental set-up. All data and code has been made available on Zenodo. We referred to the changes (i.e. lines) we made in the manuscript underneath each appropriate comment of the reviewers.*

*Kind regards,*

**By Fabrizio D'Aprile**

Certainly, water stress and/or related soil parameters cannot explain all the time variability in foliage
coloration. However, it can contribute and therefore depict reaction of vegetation at both the small parcel
and landscape level. I tested it in two works (please see the attached publications). This does not mean to
contrast the authors' work but providing them with some more information that might be useful. And, I
would also like to attract the attention on the fact that crown transparency and coloration/withering,
which have frequently been used as parameters related to water stress due to both direct and indirect
causes, may not necessarily be indicative of the level of the humidity content of the tree and/or health
status.

*Dear Mr. Aprile,*
*Thank you for your comments and articles. We will consider them for revision and future work.*
*Certainly, further studies over a more extensive geographical range could aid to further unravel*
*the effects of various water stress and/or soil parameters on the leaf coloration dynamics. Note*
*that we added data on the volumetric soil water content measured at the flux tower in*
*Brasschaat (see L. 204; fig. 2; L. 219 – 231 and L. 244 -252) and commented on this data (L. 477 –*
*479 and L. 612 - 618).*
*Kind regards,*

**By Anonymous Referee #1**

Mariën and co-authors presented an experimental study in which they assessed the effect of drought stress on leaf senescence for 3 tree species in Belgium forests (mature trees) over 2017-2019 and from manipulative experiments with saplings. Results did not show any effect of drought stress on the timing of leaf senescence. However, the authors observed an effect of drought on the chlorophyll content and the canopy greenness of trees. Overall, this study is well written. The experimental design is sound, and limits in the protocol and analysis are clearly highlighted and discussed. Results support the conclusions of the manuscript. I don't have major comments on the manuscript, only a few suggestions that might strengthen the analysis:

1) The authors used piecewise logistic regression to estimate the timing of leaf. Some studies suggested that a simple threshold approach leads to better results and maybe a more robust comparison of phenological events. Did the authors tried to compare the timing of senescence using a threshold approach? The absence of observed effect might come from the definition of leaf senescence.

2) Drought stress is defined here as the rainfall deficit. Instead of a meteorological drought index, did the authors tried other physiological drought indices? For example, the ratio of actual over potential evapotranspiration (Stocker et al. 2018) that might be more representative of the stress than the rainfall deficit.

3) Some studies suggested that the timing in leaf unfolding impacts senescence (Fu et al. 2014). Was this effect observed on site? Did the authors include other effects than precipitation, temperature and drought stress in their model? It might be interesting to discuss this point in the discussion section.

4) I suggest the authors to highlight the effect of drought stress on CCI and greenness in the abstract. As they discussed (L. 464), the effect on greenness is probably an important source of confusion in the literature and I think it is an important message of this paper. I hope the authors will find these comments useful for improving their manuscript. Best regards,

*Dear Anonymous Referee 1,*

*Thank you for your review and suggestions. We will respond here to your comments:*

1) *The Referee asks whether we considered alternatives (e.g. simple thresholds in canopy coloration percentage) to the piecewise linear regressions to determine the timing of the onset of leaf senescence. We are aware that different methodologies (e.g. from simple thresholds to complex network-based approaches) can, and are, used to estimate the timing of leaf senescence. In fact, we compared the results obtained using piecewise linear regressions and 50% canopy coloration / leaf fall thresholds (i.e. assuming that the onset of leaf senescence can be approximated with the timing when 50% of the canopy lost the green color) in previous work, and showed that the methods provide different results, with 50% thresholds giving results that are consistently later (Mariën et al., 2019). We agree that comparing different methods might nevertheless yield advantages, as the timing of leaf senescence is inherently a problem of deriving a trend in complex ecological data (i.e. data that is, for example, hierarchical and non-linear). Exactly for deriving this trend, and as extra regression method to compare to the piecewise linear regressions, we used the generalized additive mixed models*

| 98 | *and plotted the resulting factor-smooth interaction smoothers with 95% simultaneous* |
| 99 | *confidence intervals.* |
| 100 | |
| 101 | 2) *The Referee asks whether we considered different physiological drought indices (e.g. the ratio* |
| 102 | *of actual over potential evapotranspiration as in Stocker et al. (2018)). We agree that other* |
| 103 | *indices would be useful. However, calculating the index proposed in Stocker et al. (2018) would* |
| 104 | *not be feasible in a short term. An additional difficulty is that these calculations would require* |
| 105 | *a hydrological model and are strongly dependent on local soil characteristics. Furthermore,* |
| 106 | *most local meteorological stations do not provide evaporation data. Finally, note that long-* |
| 107 | *term values of the rainfall deficit, as reported in Fig. 3, are rather exceptional. Therefore, the* |
| 108 | *drought stress index that is reported here should be sufficiently representative for our* |
| 109 | *purposes. Note that we actually do not use the drought index in our calculations or models but* |
| 110 | *only use it to describe the meteorological conditions within the three year study period.* |
| 111 | *We also added data on the volumetric soil water content measured at the flux tower in* |
| 112 | *Brasschaat (see L. 204; fig. 2; L. 219 – 231 and L. 244 -252) and commented on this data (L.* |
| 113 | *477 – 479 and L. 612 - 618), which can give an additional indication of the water deficit in the* |
| 114 | *study area during the considered period.* |
| 115 | |
| 116 | *The Referee asks whether we observed an effect of the timing of the leaf unfolding on the* |
| 117 | *senescence timing, and whether we considered including other variables into our model. The* |
| 118 | *age of leaves might indeed affect the timing of the onset of senescence, especially in species* |
| 119 | *with an indeterministic growth pattern (e.g. birch). Therefore, we will test the correlation* |
| 120 | *between leaf unfolding and senescence timing (some preliminary results are available in the* |
| 121 | *supplementary file 'TEST_BB_OLS_markdown'). However, our dataset will be limited to* |
| 122 | *mature trees in 2018 and 2019, as spring data for 2017 are not available and leaf unfolding* |
| 123 | *for the trees in the manipulative experiment was affected by establishment effects. However,* |
| 124 | *note that we did not follow the exact same leaves from bud burst to senescence. In addition,* |
| 125 | *it is hard to disentangle whether the different timing of the bud burst affects the timing of leaf* |
| 126 | *senescence, or whether the opposite is the case (Marchand et al 2020). Our models simply* |
| 127 | *included "treatment", "leaf position", "day of year" and "individual_tree" for the manipulative* |
| 128 | *experiment, and "year", "species", "leaf position", "day of year" and "individual_tree" for* |
| 129 | *mature trees. So, they did not include meteorological variables. A significant upgrade in the* |
| 130 | *modeling work is the application of GAMM or GAMLSS models, where correlations between* |
| 131 | *e.g. seasonal chlorophyll data and meteorological data, are accounted for. The amount of* |
| 132 | *work in applying these models to senescence trends is significant and we are working on this* |
| 133 | *in a next manuscript.* The following line was added at L. 630 – 633. "Although Fu et al. (2014) |
| 134 | suggested a correlation between the bud burst and the onset of autumn leaf senescence, we |
| 135 | have found no relationships for 2018 and 2019 in birch and beech, and a positive relationship |
| 136 | in oak (every delay of one day in the bud burst corresponded to a delay of ± two days in the |
| 137 | onset of autumn leaf senescence)". More details about this are added in a specific file |
| 138 | uploaded in Zenodo. |
| 139 | |
| 140 | 3) *The referee suggests highlighting the effect of drought stress on CCI and greenness in the* |
| 141 | *abstract, as discussed on L. 464 ("For the mature trees, the different drought response of the* |
| 142 | *autumn pattern of chlorophyll (no effect) and the loss of canopy greenness (advanced and* |

*enhanced) is probably an important reason of confusion still present today in the literature on*

*the relationship between drought and autumn senescence"). We thank the referee for this*

*suggestion and will consider this in the revision. We highlighted the different effect of the*

*drought on the CCI and canopy greenness in the abstract (L. 30 – 33) and text (L. 597 – 605).*

Kind regards,

**By Anonymous Referee #2**

The article analyzes the impact of drought on the onset of autumn senescence and the difference featured by different temperate deciduous tree species. The authors used a manipulative experiment of tree beech sapling and three years of data on beech, birch, and oak trees. The authors show that drought did not affect the onset of senescence. Tree saplings showed high mortality with drought, and mature trees showed higher leaf mortality. No significant differences across species were observed. The manuscript deals with a significant subject, senescence, about which not much is yet known. Understanding the senescence process, particularly in relation to drought, is fundamental to predict the phenological cycle of temperate trees better.

1) Regarding the greenhouse experiment, I have a methodological concern. From the data reported in Fig

1 seems that the "drought" treatment does not have a significant (should be tested statistically thought)

effect on soil moisture (Fig 3c). Instead, the effect was mainly an increase of VPD that is not drought but an increase of the atmospheric evaporative demand. One of the factors linked to the earlier senescence in the case of drought is abscisic acid accumulation (ABA). Long term ABA responses should be more induced by soil moisture. Root perceives reducing soil moisture and upregulate ABA synthesis. ABA is a factor controlling earlier senescence (and stomatal regulation). I am not aware of studies showing the high VPD can trigger the same response in terms of upregulation of ABA. It could be that the lack of response observed was simply due to the fact that the reduction of soil moisture was not enough to trigger the physiological response inducing earlier senescence.

2) Also, in general, I would not call drought the treatment. Given the data shown in Figure 1, I think it is more heat stress. Please provide more insights to understand whether the treatment can be indeed called drought treatment. If not, I would suggest talking about heat stress and increased atmospheric aridity.

This would not diminish the paper. There is a lot of discussion on the different repose of plants to decreasing soil moisture and/or increasing VPD, and here I think the authors are looking at increased VPD

and not necessarily at drought. This can also support the discussion of the differences between 2018

(more soil moisture stress) and 2019 (more heat and VDP stress) – see discussion at line 469-470.

3) The 20% reduction of incoming light should also be better addressed (Line 162-163): though unclear, it seems that senescence is controlled by photoperiod. How does 20% - decrease in incoming radiation affect the photoperiod? The authors should check this and evaluate if the reduction of light has an impact on the results.

4) In the methods section 2.1.2, when the CCI is mentioned the first time, I expected a description of the sampling (that comes later). I think it would be beneficial to move section 2.2 above, where the CCI is mentioned the first time. Preferentially, put a reference in paragraph 2.1.2 to paragraph 2.2.

5) The meteo stations are 20 and 60 km from the sites. But there is no information about where these stations were located (in a city, in a forest, in a grassland, at which height). Even if the climate regimes can be similar at a distance of 20-60 km can we be sure that the microclimate is comparable? I suggest the authors carefully check all this information and provide a methods description that can prove the study's robustness.

6) The equation and symbols do not follow the scientific format. I suggest to rewrite them. Also many variables have names that are more for a programming language but not following the scientific notion. I suggest to follow the IUPAC standards, or at least try to go close to that format. Avoid using "Leaf_place"Also please define the variable the first time is used, and then stick with the symbol: one example is the "day of the year" that in the equation 2 (model 1) is Doy and in the text is "day of the year"

7) If I am not wrong there is a mistake in Eq 1. First if rH should not be expressed in % as indicated but as fraction (rH[%]/100) Here the result of VPD with the current equation > T <- 25 > rH <- 50 > e0 <- 613.75*exp((17.502*T)/(240.97+T)) > e <- rH*e0 > VPD <- e0-e > VPD [1] -155829.6 Moreover, even if the rH is used in the correct unit, the VPD unit is wrong. The resulting VPD from this equation is in Pa and not kPa as indicated at line 144. > T <- 25 > rH <- 50/100 > e0 <- 613.75*exp((17.502*T)/(240.97+T)) > e <- rH*e0 > VPD <- e0-e > VPD [1] 1590.098 The VPD reported in the figures seems correct, therefore please verify is there is a problem in the Equation.

8) There are few track changes and typos in the manuscript. Please edit careful the article a. Line 249, line 415, 416, 417, 4)

9) The reference to the R package is a bit strange R/ggpubr etc. Please modify in: "we use the R package ggpubr (Reference)". But it is very nice that the authors cite all the packages. This is important and often overlooked.

10) Please report "Model 1 and 2" in a less R script style. Please use mathematical notation

11) I think the breakpoint analysis was achieved with the "segmented" package and not dplyr", correct?

12) Line 464-465 – this is interesting, please elaborate more this point.

*Dear Anonymous Referee 2,*

*Thank you for your review and suggestions. We will respond here to your comments:*

1) *The Referee asks whether it is possible that the reduction of soil moisture in the glasshouse experiment was not enough to trigger the physiological response inducing earlier senescence. He therefore questions whether an increased VPD can trigger the upregulation of ABA. Literature shows that ABA, which is known to control earlier senescence, is indeed upregulated as a response to the stomatal changes corresponding to changing vapor pressure deficit levels (McAdam and Brodribb, 2016;McAdam et al., 2016;Bauerle et al., 2004;Xie et al., 2006). We agree that the treatments +0˚C and +3˚C did not result in large differences in soil water content. However, we will test this statistically, as suggested (for example, see the supplementary file 'TEST_SWC_markdown' and Rose et al. (2012) for additional information on the possible statistical methodology). On the other hand, it is likely that larger differences were present between the reference plots and the treatments, as the reference plots were*

*irrigated more (L. 160), and such irrigation regime showed values of soil water content of up*
*to ca. 0.25 m³/m while the values of 0.05 m³/m³ were reached in the treatments (see Fig 1;*
*unfortunately, sensor malfunctioning did not allow us to gather soil water content data for*
*the reference plots). Given that we observed a high mortality, it might have been the case that*
*our +3 °C treatment was too extreme, triggering necrosis instead of earlier senescence*
*(Munné-Bosch and Alegre, 2004). We have commented extensively on the different*
*interpretation of the treatment effects and the lack of soil water content measurements in the*
*reference plots in L. 169 – 179.*

2)  *The Referee suggests we talk about heat stress rather than drought stress. As mentioned*
*above, the reference plots were irrigated more than the treatments plots (L. 148 - 149; 159 -*
*160). Therefore, the more appropriate definition would be "treatment based on warming, less*
*irrigation and increased atmospheric aridity". We could use this definition (although longer*
*and somewhat impractical it is the closest to reality). In reference to L. 469 - 470 ("…the*
*drought of 2019, which coincided with several heat waves, might have been less damaging for*
*late summer leaf dynamics, than the drought of 2018…"), a more detailed comparison*
*between experimental manipulation and mature trees in years 2018 and 2019 would have*
*required a factorial approach separating drought and warming, while our design was more*
*basic. In addition, as shown in figure 3, the rainfall deficit was high in all years. It is true that*
*the rainfall deficit was extremely high in 2018 – 2019, but the rainfall deficit was also high in*
*2017 – 2018 and 2019 – 2020. Likely, more site specific measurements on the soil water*
*content would indeed have been useful. Note, however, that figure 2 and table 1 also indicate*
*that there was not only little precipitation but also that this precipitation fell in irregular*
*patterns, making potential droughts more likely. We have commented on the interpretation*
*of the treatment effect in L. 171 – 179. In addition, we have adapted our terminology to heat*
*stress and increased atmospheric aridity were appropriate (e.g. see L. 178; L. 491; L. 580; L.*
*585; L. 588: L. 620; L. 662 and L 691 – 693). Finally, we improved the discussion about the*
*treatments (L. 589 - 594). We also added data on the volumetric soil water content measured*
*at the flux tower in Brasschaat (see L. 204; fig. 2; L. 219 – 231 and L. 244 -252) and commented*
*on this data (L. 477 – 479 and L. 612 - 618).*

3)  *The Referee asks to comment on the effect of the 20% reduction in light due to the colorless*
*polycarbonate roof in the glasshouses (L. 162 – 163; "A draw-back of the experiment is that*
*the saplings in the reference plots received more incoming light (i.e. ± 20%) than the saplings*
*in the glasshouses (Van den Berge et al., 2011)"). The Reviewer raises an interesting point: can*
*a reduction / change in the light affect the photoperiod? Preliminary tests suggested that the*
*ratio of light in different wavelengths (e.g. R/FR) during civil twilight (i.e. what is required for*
*phytochrome to detect the photoperiod) does not change seasonally significantly in our study*
*area. This provide indirect evidence for us to believe that our light reduction (limited to 20%),*
*combined with the fact that very low light intensities are needed for plants to detect*
*photoperiod (Legris et al., 2019;Poorter et al., 2019;Franklin and Quail, 2010), would not have*
*caused significant changes in photoperiod. We agree that it could be interesting to test the*
*effect of the roof alone. However, this is not feasible in the short term. The effect of the roof*
*is also partly captured by the results on the saplings in the +0 °C treatment glasshouses. We*

*have added extensive comments on the (potential) effect of the polycarbonate roof in L. 125*
*– 126; L. 184 – 189 and L. 650 – 659)*
4)   *The Referee asks to consider restructuring section 2.2 and 2.1.2. We will consider this in the*
*revision. This part has been improved. For clarity, measurements of CCI and loss of canopy*
*greenness are not mentioned anymore when describing sites and climate (2.1.2) but only later*
*on (in 2.2).*
5)   *The Referee asks to provide more information on the meteorological stations. We will add the*
*following information to the manuscript in the revision. (1) The station of Ukkel is located*
*within a green area in the suburb of Brussels (thus, classifiable as "urban park"). The*
*microclimate is expected to be different than at our study sites. However, data from Ukkel*
*were used to describe the intra-annual variability and long-term trends (Table 1 and Fig. 3),*
*which are less affected by microclimate. (2) The meteorological station of Brasschaat is very*
*close to our sampling site in the Park of Brasschaat and in the Klein Schietveld (± 3 km and ± 4*
*km, respectively). The meteorological station in Brasschaat is a 40 m high scaffolding tower,*
*at which measurements are taken at various heights, and stands in a patch of mixed forest*
*covered mainly by Scots pines and deciduous tree species, such as oak and birch (see Carrara*
*et al. (2003) for more information). Data of the temperature, precipitation and humidity were*
*taken at the top of the tower. Data from Brasschaat were used to describe the seasonal*
*pattern in 2017, 2018 and 2019, and as input to the models. (3) The station of Woensdrecht is*
*located in an open field at a local airport surrounded by heathland and urban area. It is located*
*near the Markiezaatsmeer, an enclosed swamp ecosystem, within the river mouth of the*
*Schelde. The measurements in both Ukkel and Woensdrecht are taken at a height of 1.5 m.*
*However, these data were only used as gap-filling in case of short term gaps in the long-term*
*Brasschaat series. In terms of differences in the microclimate, it is indeed not ideal that we*
*needed to use data from the meteorological stations of Ukkel and Woensdrecht. However, we*
*are limited here by the availability of the data and the meteorological stations of Ukkel and*
*Woensdrecht are closest (and most representative) for our sampling sites. Note that we added*
*this (and more) information in L. 233 – 255.*
6)   *The Referee comments on the style of the model notation and suggests to better define the*
*variables at first use. We will define the variables further at first use and avoid inconsistencies.*
*However, both the descriptive style and mathematical notation are based on examples and*
*suggested notation in the specific literature (Zuur et al., 2007;Zuur et al., 2010;Zuur et al.,*
*2011;Zuur et al., 2016;Simpson, 2018;Pedersen et al., 2019;Wood, 2017) and readers*
*interested in background references might find it easier if style consistency is respected.*
*Perhaps the Editor can comment on the journals preference? We have removed the*
*abbreviations and inconsistencies for the explanatory variables in the model notations (see L.*
*345; L. 356; L. 369 and L.380). Some notation is not significantly changed because it follows*
*similar literature and because there was no agreement on the alternative.*
7)   *The Referee notes there is an error in the units of the equation on the vapor pressure deficit.*
*Thanks, we will correct this in the revision. The actual and saturation vapor pressure deficit*

*are indeed in Pa, while the relative humidity should be noted as a fraction. The data was indeed calculated using the correct equation. We have corrected the equation and the kPa to Pa (see L. 142 – 148).*

8) *The Referee points out some typo's. The will be addressed in the revision. All typos have been changed (e.g. see L. 316; L. 503- 505; …)*

9) *The Referee suggests to write R packages in a different format. If preferred by the Editor, we will address this in the revision. The editor considered this optional.*

10) *The Referee suggests using only the mathematical notation for model 1 and 2.Considering the literature (see the response on comment 6) and the preference of the Editor, we will address this in the revision. The editor considered this optional.*

11) *The Referee suggests to remove the reference to the R package "DPLYR" as the breakpoint analysis is done only using the R package "SEGMENTED". While "DPLYR" was used for data wrangling, we agree "SEGMENTED" is indeed the package that is used for the breakpoint analysis. We will remove the reference to "DPLYR" in the revision. We removed the reference to the use of this package (see L. 405 – 406).*

12) *The referee asks to elaborate on L. 464. ("For the mature trees, the different drought response of the autumn pattern of chlorophyll (no effect) and the loss of canopy greenness (advanced and enhanced) is probably an important reason of confusion still present today in the literature on the relationship between drought and autumn senescence"). We thank the referee for this suggestion and will consider this in the revision. While the detoxification of chlorophyll is a prerequisite for the expression of different coloration values, chlorophyll does not degrade at the same speed as other leaf pigments. In fact, not even all leaf pigments degrade (or are formed) at the same velocity throughout the senescence process (Keskitalo et al., 2005). Consequently, observations of changing coloration levels are difficult to interpret. Moreover, note that coloration measurements also take into account leaf yellowing and mortality due to hydraulic failure. We elaborated on the different effect of the drought on the CCI and canopy greenness in the abstract (L. 30 – 33) and text (L. 597 – 605).*

*Kind regards,*

**References:**

Bauerle, W., Whitlow, T., Setter, T., and Vermeylen, F.: Abscisic acid synthesis in Acer rubrum L. leaves—a vapor-pressure-deficit-mediated response, J. Am. Soc. Hort. Sci., 129, 182-187, 2004.

Carrara, A., Kowalski, A. S., Neirynck, J., Janssens, I. A., Yuste, J. C., and Ceulemans, R.: Net ecosystem $CO_2$ exchange of mixed forest in Belgium over 5 years, Agricultural and Forest Meteorology, 119, 209-227, https://doi.org/10.1016/S0168-1923(03)00120-5, 2003.

Franklin, K. A., and Quail, P. H.: Phytochrome functions in Arabidopsis development, J. Exp. Bot., 61 1, 11-24, 2010.

Keskitalo, J., Bergquist, G., Gardestrom, P., and Jansson, S.: A cellular timetable of autumn senescence, Plant Physiol., 139, 1635-1648, 10.1104/pp.105.066845, 2005.

Legris, M., Ince, Y., and Fankhauser, C.: Molecular mechanisms underlying phytochrome-controlled morphogenesis in plants, Nat Commun, 10, 5219, 10.1038/s41467-019-13045-0, 2019.

Mariën, B., Balzarolo, M., Dox, I., Leys, S., Lorene, M. J., Geron, C., Portillo-Estrada, M., AbdElgawad, H., Asard, H., and Campioli, M.: Detecting the onset of autumn leaf senescence in deciduous forest trees of the temperate zone, New Phytol, 224, 166-176, 10.1111/nph.15991, 2019.

McAdam, S. A., and Brodribb, T. J.: Linking Turgor with ABA Biosynthesis: Implications for Stomatal Responses to Vapor Pressure Deficit across Land Plants, Plant Physiol., 171, 2008-2016, 10.1104/pp.16.00380, 2016.

McAdam, S. A., Sussmilch, F. C., and Brodribb, T. J.: Stomatal responses to vapour pressure deficit are regulated by high speed gene expression in angiosperms, Plant, Cell Environ., 39, 485-491, 2016.

Munné-Bosch, S., and Alegre, L.: Die and let live: leaf senescence contributes to plant survival under drought stress, Funct. Plant Biol., 31, 10.1071/fp03236, 2004.

Pedersen, E. J., Miller, D. L., Simpson, G. L., and Ross, N.: Hierarchical generalized additive models in ecology: an introduction with mgcv, PeerJ, 7, e6876, 10.7717/peerj.6876, 2019.

Poorter, H., Niinemets, Ü., Ntagkas, N., Siebenkäs, A., Mäenpää, M., Matsubara, S., and Pons, T.: A meta-analysis of plant responses to light intensity for 70 traits ranging from molecules to whole plant performance, New Phytol., 223, 1073-1105, https://doi.org/10.1111/nph.15754, 2019.

Rose, N. L., Yang, H., Turner, S. D., and Simpson, G. L.: An assessment of the mechanisms for the transfer of lead and mercury from atmospherically contaminated organic soils to lake sediments with particular reference to Scotland, UK, Geochim. Cosmochim. Acta, 82, 113-135, 10.1016/j.gca.2010.12.026, 2012.

Simpson, G. L.: Modelling Palaeoecological Time Series Using Generalised Additive Models, Frontiers in Ecology and Evolution, 6, ARTN 149

10.3389/fevo.2018.00149, 2018.

Stocker, B. D., Zscheischler, J., Keenan, T. F., Prentice, I. C., Peñuelas, J., and Seneviratne, S. I.: Quantifying soil moisture impacts on light use efficiency across biomes, New Phytol., 218, 1430-1449, https://doi.org/10.1111/nph.15123, 2018.

Wood, S.: Generalized additive Models, Chapman and Hall/CRC, New York, 2017.

Xie, X., Wang, Y., Williamson, L., Holroyd, G. H., Tagliavia, C., Murchie, E., Theobald, J., Knight, M. R., Davies, W. J., and Leyser, H. O.: The identification of genes involved in the stomatal response to reduced atmospheric relative humidity, Curr. Biol., 16, 882-887, 2006.

Zuur, A., Ieno, E., and Smith, G.: Analysing Ecological Data, Statistics for Biology and Health, 2007.

Zuur, A. F., Ieno, E. N., and Elphick, C. S.: A protocol for data exploration to avoid common statistical problems, Methods Ecol. Evol., 1, 3-14, 10.1111/j.2041-210X.2009.00001.x, 2010.

Zuur, A. F., Ieno, E. N., Walker, N. J., Saveliev, A. A., and Smith, G. M.: Mixed Effects Models and Extensions in Ecology with R, Springer, 2011.

Zuur, A. F., Ieno, E. N., and Freckleton, R.: A protocol for conducting and presenting results of regression-type analyses, Methods Ecol. Evol., 7, 636-645, 10.1111/2041-210x.12577, 2016.

---

## Author Response (AR2)

**Note:** Comments by the Editor and Reviewers on the major review are presented in red, while responses (i.e. lines of changes that have been made in the manuscript) of the authors are given in blue, respectively.

**By the Editor**

Thanks for your careful and comprehensive responses and revisions to the first-round reviews. Both
reviewers have assessed the revised manuscript and find it much improved: clearer, strengthened in
almost every respect, and stronger. Both now recommend acceptance, although R2 still has a number of
minor                    technical                    suggestions                    and                    questions.
I have read the manuscript and agree with the reviewers. After addressing R2's remaining comments--
which should not take much work--I think this will be fully acceptable for final publication in
Biogeosciences. Congratulations on an interesting and well-done study.

*Dear Editor,*
*Thank you for your suggestions and comments. We have changed or answered the additional*
*comments of Referee 2.*
*Kind regards,*

**By Anonymous Referee #1**

The authors made a nice work in revising their manuscript. They clearly justified and discussed their choices, as well as the results and limits of their work. I have no further comments on their manuscript.

*Dear Anonymous Referee 1,*

*Thank you for your kind words and review.*

Kind regards,

**By Anonymous Referee #2**

Dear Authors,

Thank you for your answer and work on the manuscript. I think the manuscript improved a lot compared to the previous version. Please find below few additional clarifications in my opinion needed before acceptance

1) Lines 81-82: Despite its relevance, literature on autumn senescence has maintained a wide variety of definitions and observational methods (Gill et al., 2015;Fracheboud et al., 2009;Gallinat et al., 2015). Please clarify what you exactly mean with this statement. I think is meant that there isn't too much clarity in the definition of senescence and on the observational methods.

2) Figure 1: Solid lines represent regressions of half-hourly measurements of the relative humidity. I do not completely understand what does it mean, you mean that the points were interpolated? Also the lines seems extremely smooth, therefore please clarify how the smoothing was done. Why not reporting the original data?

3) Lines 178-179 "significantly lower (P < 2 x 10-16) in comparison 178 to the glasshouses with the +0 °C treatment". Please add which test was conducted to assess this and use p <0.001 to indicate highly significance (P < 2 x 10-16 is extremely unusual).

4) Line 180: Please indicate in between which treatment you have the difference. Wasn't the soil moisture not available for reference plot? This was indicated at line 172-174 (please clarify).

5) Caption Table 1: "The degree of abnormality of the values is represented by (a; abnormal values that happen on average once every 6 years) and (e; exceptional values that happen on average once every thirty years). Since 2019, the KMI uses a new system to show the degree of abnormality." Please clarify what is abnormality, how is calculated, it is not really clear what that table is showing.

6) Line 244: Interannual variability – specify of meteorological variables or driver

7) Line 328: The model assumptions were tested following Zuur et al. (2010) and using R/ggpubr (Kassambara, 2019). I am confused, ggpubr is a package mostly for vizualization and few features for data analysis.

8) Line 336: "and R/DPLYR". This was the same comment in the previous revision. Dplyr is a package for data manipulation and for sure does not contain any function to fit GAMM. I guess the author should refer only to mgcv r other packages but not dplyr.

9) Table 2: I would include the AIC and remove the equation (and keeping only the Model number as the equation is already reported in the text)

10) Line 419: "Trees that did not show a clear breakpoint (13 in the manipulative experiment) were not considered in the analysis". Isn't this a bit critical? I think they should be kept in the analysis . Or perhaps the authors can try to bootstrap the dataset and identify for each tree the breakpoint and the uncertainty, and then use an objective criteria to exclude some of trees

11) Paragraph 3.2: I think the authors should also describe the big differences in CCI in the +3 C treatment better. I cannot see a description of that interesting behavior in the result section (unless I missed it). Also I think that in Fig 4 the GAMM make up some of the differences between the treatments toward the end of the seasonal development (e.g. Fig 4A particularly for Fig 4A +3C treatment). The authors do not over interpret this issue but I think they should first check that the parameter of the smoothing functions in the GAMM is correctly set, and second mention this aspect in the results.

12) The discussion section is very nice. Still I think that some of the statement should be re-discussed after some of the concerns raised above are clarified.

*Dear Anonymous Referee 2,*

*Thank you for your review and additional suggestions. We will respond here to your comments:*

1.  *The Referee asks whether L. 81-82 (Despite its relevance, literature on autumn senescence has*
*maintained a wide variety of definitions and observational methods (Gill et al., 2015;Fracheboud*
*et al., 2009;Gallinat et al., 2015) means that there isn't too much clarity in the definition of*
*senescence and on the observational methods. This is indeed how this sentence should be*
*interpreted. We have clarified this sentence to avoid confusion. Now it reads as follow: "Literature*
*reports several definitions of autumn senescence and of multiple observational methods to*
*measure autumn senescence (Gill et al., 2015;Fracheboud et al., 2009;Gallinat et al., 2015)." (L.*
*80 - 81).*

2)  *The Referee asks whether we mean with "Solid lines represent regressions of half-hourly*
*measurements of the relative humidity" that the points concerning the relative humidity in figure*
*1 were interpolated. The Referee also asks how the smoothing was done and why the original data*
*is not reported instead. The points concerning the relative humidity in figure 1 were indeed*
*interpolated using the geom_smooth argument from the R/ggplot2 package. Because, the*
*geom_smooth arguments used GAMs for the interpolation they represent in fact regressions (info*
*on this is now reported in the legend of Figure 1; L. 154 - 155). Given that the data is measured*
*every half-hour, with the logical differences throughout the day, we chose to represent the trend*
*here rather than the original data. This would have made the graph less clear. Original data of*
*relative humidity are available at Zenodo doi:* 10.5281/zenodo.4559535.

3)  *The Referee asks to specify the test that was used to support the statement in L. 178 – 179*
*("significantly lower (P < 2 x 10-16) in comparison 178 to the glasshouses with the +0 °C*
*treatment"). The Referee also suggest to use p < 0.001 to indicate the high significance. This p-*
*value is derived from the application of GAMMs, as described in the supplementary file*
*'Test_SWC'. This file was made in response to a previous suggestion concerning the lack of*
*statistical testing of the difference in the soil water content results and it is available for readers*
*(see Data availability Zenodo doi:* 10.5281/zenodo.4559535 *to find the document). In the text, we*
*have changed the value to p < 0.001 and now refer the reader to the Data availability section for*
*more information (L. 175 179).*

4)  *The Referee asks to indicate the treatments in L. 180 to which the mentioned difference refers to.*
*The Referee also asks whether the soil moisture data was available for the reference plot. The*
*Referee suggests to make a connection to L. 172 – 174. We have specified in the text that this*
*difference refers to the +0 °C and +3 °C treatments (L. 177 - 179). Indeed, soil moisture data was*
*not available for the reference plots.*

*5) The Referee asks to clarify further what the abnormality means in the caption of Table 1 ("The*
*degree of abnormality of the values is represented by (a; abnormal values that happen on average*
*once every 6 years) and (e; exceptional values that happen on average once every thirty years").*
*The Referee also asks how this is calculated and what the Table actually shows. Table 1 represents*
*the meteorological conditions in summer and autumn in Ukkel, Belgium. It indicates which average*
*values are considered to be normal (given the reference period 1981 – 2010) and how the average*
*values measured in 2017, 2018 and 2019 compare to these normal values. Whenever a value*
*measured in 2017, 2018 or 2019 is extremely high/low in comparison to the reference period, this*
*would be indicated using a label of abnormality (e.g. abnormal, exceptional or within the highest*
*three values recorded). The description of the label of abnormality is now more clearly reported in*
*caption of Table 1 at L. 215 -220 ("The degree of abnormality of the values is represented by two*
*labels: a for abnormal values (with a recurrence time of six years) or e for exceptional values (with*
*a recurrence time of thirty years). In case only one month had abnormal values, this label is*
*followed by the name of that particular month. Since 2019, the KMI uses a new system to show*
*the degree of abnormality: values that are with the five highest values since 1981 are marked by*
*(+), while values within the three highest values are marked by (++)."). Consider for example the*
*total precipitation in summer, here an 'abnormal' value (e.g. 134.7 mm in 2018) has a recurrence*
*time of six years during the reference period 1981 – 2010. Note that the reported system to show*
*abnormality of values is the standard of the Belgian Royal Meteorological service (KMI) for these*
*years. The main purpose of this table is to give an indication of the normal meteorological*
*conditions at our sites, and how the heat and drought stress measured during 2017, 2018 and*
*2019 compare to these normal values.*

*6) The Referee asks to specify in L. 244 whether the inter-annual variability refers to meteorological*
*variables or drivers. We have specified in the text that the inter-annual variability and long-term*
*trends refers to the meteorological variables (L. 242).*

*7) The Referee asks why the package R/ggpubr is used for data analysis, and suggests to mention it*
*for data visualization instead. We agree. The package is now reported among the others*
*visualization packages (L. 322 – 324).*

*8) The Referee suggests to mention only the package R/mgcv in L.336 since the package R/dplyr is*
*used for data manipulation. We agree that R/DPLYR cannot be used to build the GAMMs. We have*
*clarified that R/DPLYR was used for data manipulation instead, together with the other more*
*general packages (L. 322 – 324).*

*9) Concerning Table 2, the Referee suggests to add the AIC of the models and to remove the*
*equations. We have added the AIC of the models. However, we choose not to remove the equations*
*to have a completed summary table, independent from the text (note that equations do not take*
*much space; L. 399).*

10) *The Referee asks whether the removal of 13 saplings from the dataset because they didn't show a clear breakpoint isn't too critical (see L. 419 "Trees that did not show a clear breakpoint (13 in the manipulative experiment) were not considered in the analysis"). He suggests to use bootstrapping.*
*Finding a significant shift (be it by means of a breakpoint, changepoints, inflection point, et cet.) is sometimes difficult and prone to limitations. Certainly, breakpoints only make sense in the case the trend can be divided in minimum two distinguishable linear trends. This is not always the case (e.g. when the overall trend tends towards a linear trend). Alternative methods to assess shifts in a trend with uncertainty are under consideration in relation to our dataset. However, the complexity of the regressions (e.g. GAMLSS + bootstrapping of additive models) required for this kind of methodology does not suit the purpose of this manuscript and it will be covered in future planned work. The 13 saplings that did not show a clear breakpoints mainly lacked elasticity in their trend, making it impossible to calculate breakpoints. However, the data of these 13 saplings is considered in the GAMMs and line plots represented in the article (Fig. 4). Figure 4 therefore gives an accurate representation of the effect of data from these 13 saplings as well.*

11) *The Referee asks to describe the big differences in the CCI of the +3 °C treatment better in paragraph 3.2. The Referee likes to see a clearer description of this behavior in the result section. The Referee points out that the GAMM's seem to make up some differences between the treatments towards the end of the seasonal development (see Fig 4A the +3 °C treatment in particular). While the Referee admits that we do not over interpret this issue, the Referee would nevertheless like us to check the parameter settings of the smoothing functions and to mention this behavior in the results sections.*
*The behavior of the CCI trend of the +3 °C treatment is described in section 3.2. However, we have highlighted this behavior further in the Discussion (L. 602 – 604; "The decline in the CCI of the saplings exposed to the +3°C treatment, around mid-August, might indicate that physiological damage due to stress can accumulate and become apparent even though stress is alleviated."). Concerning, the modelling issue for the + 3°C treatment at the end of the season, we added a line in the Results (L. 504 – 505; "From the end of September, the CCI decreased in all treatments, showing similar CCI measurements across treatments. However, the modeled CCI of the +3 °C treatment declined slower than the modeled CCI of the other two treatments."). However, we also commented on this in the Discussion (L. 698 – 702; "Overall, the GAMMs reproduced reliable fits of the CCI and canopy greenness. One of the few observed issues was a small mismatch between the mean CCI shown by the smoother of the fitted GAMM and the mean CCI shown by the line plot for the + 3°C treatment at the end of the growing season (early October – mid November). The overestimation of the CCI in this case might reflect the limitations of using Gaussian GAMMs here."). Note that, due to the factor-smooth interaction smoother, the smoothing functions had the same parameter settings across the treatments. The Referee correctly points out that this behavior should not be over interpreted.*

12) *The Referee thinks some of the statements in the discussion might need reconsideration after implementing some of the new suggestions.*

*The Discussion was amended, particularly by addressing point 11. Moreover, a few minor text*
*mistakes were also corrected.*
*Kind regards,*